# Sample Complexity and Representation Ability of Test-time Scaling Paradigms

**Baihe Huang**[1], **Shanda Li**[2], **Tianhao Wu**[1], **Yiming Yang**[2], **Ameet Talwalkar**[2,3],
**Kannan Ramchandran**[1], **Michael I. Jordan**[1,4], **Jiantao Jiao**[1,5]

[1]Dept. of Electrical Engineering and Computer Sciences, University of California, Berkeley
[2]School of Computer Science, Carnegie Mellon University   [3]Datadog   [4]Inria Paris   [5]NVIDIA

Correspondence to: `baihe_huang@berkeley.edu`, `shandal@cs.cmu.edu`.

## Abstract

Test-time scaling paradigms have significantly advanced the capabilities of large language models (LLMs) on complex tasks. Despite their empirical success, theoretical understanding of the sample efficiency of various test-time strategies—such as self-consistency, best-of-$n$, and self-correction—remains limited. In this work, we first establish a separation result between two repeated sampling strategies: self-consistency requires $\Theta(1/\Delta^2)$ samples to produce the correct answer, while best-of-$n$ only needs $\Theta(1/\Delta)$, where $\Delta < 1$ denotes the probability gap between the correct and second most likely answers. Next, we present an expressiveness result for the self-correction approach with verifier feedback: it enables Transformers to simulate online learning over a pool of experts at test time. Therefore, a single Transformer architecture can provably solve multiple tasks without prior knowledge of the specific task associated with a user query, extending the representation theory of Transformers from single-task to multi-task settings. Finally, we empirically validate our theoretical results, demonstrating the practical effectiveness of self-correction methods.

## 1 Introduction

Over the past several years, Large Language Models (LLMs) have witnessed remarkable advances, achieving unprecedented performance across a broad spectrum of applications (Brown et al., 2020; Bubeck et al., 2023; Chowdhery et al., 2022). Driven by the paradigm of chain-of-thought (CoT) reasoning (Wei et al., 2022b), the outputs of LLMs have not only grown in length but also in structural complexity. In particular, recent studies have demonstrated that scaling up computational resources during test time significantly enhances the problem-solving capabilities of LLMs—a phenomenon known as test-time scaling (Brown et al., 2024; Wu et al., 2024; DeepSeek-AI, 2025; OpenAI, 2025). Various methods have been proposed to effectively utilize additional test-time compute, including self-consistency (Wang et al., 2023; Brown et al., 2024; Nguyen et al., 2024; Chen et al., 2024b), best-of-$n$ sampling (Irvine et al., 2023; Song et al., 2024a; Munkhbat et al., 2025; Qiu et al., 2024; Sessa et al., 2024), Monte Carlo Tree Search (MCTS) (Tian et al., 2024; Zhang et al., 2024d; Gao et al., 2024; Wan et al., 2024; Chen et al., 2024a; Lin et al., 2025), and self-correction (Madaan et al., 2023; Welleck et al., 2023; Chen et al., 2024d; Gou et al., 2024; Zhang et al., 2024c; Kumar et al., 2024). Powered by test-time scaling paradigms, several reasoning models, such as OpenAI-o1 (OpenAI, 2024) and DeepSeek-R1 (DeepSeek-AI, 2025), have achieved remarkable success in many complex tasks (Cobbe et al., 2021; Hendrycks et al., 2021; Shi et al., 2024; Huang et al., 2024b; Zhang et al., 2024a).

Despite these empirical advancements, the theoretical foundations of test-time scaling remain underdeveloped. While recent progress has been made in understanding the expressiveness and learnability of chain-of-thought reasoning (Feng et al., 2023; Merrill & Sabharwal, 2023; Li et al., 2024b; Joshi et al., 2025), two fundamental challenges remain unresolved:

1. Many test-time scaling approaches rely on repeated sampling from the same LLM to select a final answer (Wang et al., 2023; Brown et al., 2024; Irvine et al., 2023; Song et al., 2024a;

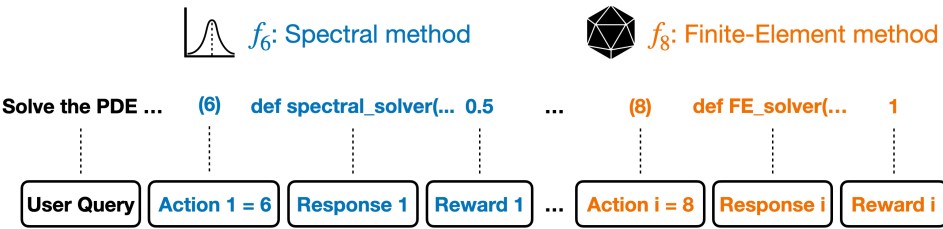

Figure 1: An illustration of test-time online learning (figure adapted from (Li et al., 2025)), where the Transformer progressively learns that finite-element method solves the partial differential equation with higher accuracy.

Nguyen et al., 2024; Chen et al., 2024b; Wu et al., 2025b; Team, 2025; Munkhbat et al., 2025; Qiu et al., 2024; Sessa et al., 2024). Two dominant paradigms are: self-consistency, which marginalizes reasoning paths and selects the most frequent answer; and best-of-$n$, which chooses the answer with the highest reward score. However, a rigorous understanding of their sample complexities is lacking. This raises the first question:

*What is the sample complexity of repeated sampling methods,*
*particularly self-consistency and best-of-$n$?*

2. Theoretical analyses of Transformers' expressiveness have largely focused on their ability to represent individual tasks (Dehghani et al., 2018; Yun et al., 2020), while their ability to express multiple tasks at the same time is under-studied. In contrast, practical LLMs have shown remarkable success in adapting to new, diverse tasks at test time via self-correction (Madaan et al., 2023; Welleck et al., 2023; Chen et al., 2024d; Gou et al., 2024; Zhang et al., 2024c; Kumar et al., 2024; Guo et al., 2025) or In-Context Reinforcement Learning (Monea et al., 2024; Tooliense, 2025; Song et al., 2025; Novikov et al., 2025). This gap underscores a need to understand how Transformers express multiple task-solving capabilities inherently and learn to adapt their behavior to the right task via test-time interaction. Therefore, we are motivated to pose the second central question:

*How can we characterize the expressiveness under test-time scaling methods,*
*especially in multi-task settings?*

**Our Contributions.** This work addresses the challenges outlined above through two key contributions. First, we analyze the sample complexity of two prominent decoding strategies: self-consistency and best-of-$n$, in terms of the *probability gap* between the most likely (correct) and the second most likely model outputs. Our results reveal a fundamental separation in sample efficiency that highlights the advantage of the best-of-$n$ approach.

**Proposition 1.1** (Informal statement of Theorem 3.1 and Theorem 3.2)**.** *Let $\Delta \in (0, 1)$ denote the difference between the Transformer's probability of producing the correct answer and the probability of the second most likely answer. Then, self-consistency requires $\Theta(1/\Delta^2)$ samples to reliably produce the correct answer, whereas best-of-$n$ achieves the same with only $\Theta(1/\Delta)$ samples.*

Second, we investigate Transformer's capacity for self-correction. We demonstrate that a Transformer equipped with verifier feedback at test time can implement online learning algorithms over a pool of expert models, enabling it to adaptively identify the most suitable expert and ultimately generate a response that maximizes the reward. This process is illustrated in Figure 1: given the user query (e.g. solve the PDE $\frac{1}{c(x)^2}\frac{\partial^2 u}{\partial t^2} - \Delta u = 0$ in $\Omega \times (0, T)$ with some boundary conditions), the Transformer $f$ autoregressively generates a sequence of actions (e.g., selecting the sixth expert) and responses (e.g., constructing and applying a spectral method solver), conditioned on the history of previous action-response pairs and their corresponding rewards (e.g., negative solution error). Notably, this process relies solely on the Transformer $f$—whose architecture encapsulates the capabilities of all experts—and the reward function, distinguishing it from traditional routing algorithms that explicitly query experts. As such, this mechanism allows a single Transformer architecture to solve multiple tasks without prior knowledge of the specific task associated with a user query.

**Proposition 1.2** (Informal statement of Theorem 4.7). *There exists a generic way to construct a wider transformer $f$ from any Transformer-based expert models $f_1, \ldots, f_K$ such that, for any task solvable by the base experts, $f$ can solve this task by self-correction which generates a sequence of responses with regret $o(1)$.*

Proposition 1.2 has two key implications. First, it demonstrates that a Transformer can express multiple tasks within a single architecture, extending beyond prior theoretical results that focus on single-task expressiveness. Importantly, the construction is task-agnostic and independent of the specific expert Transformers used, making both the result and the underlying techniques of independent theoretical interest. Second, Proposition 1.2 reveals a fundamental distinction between self-correction and repeated-sampling paradigms. While repeated-sampling methods generate identically distributed responses across attempts, self-correction *provably* allows the model to update its attempts based on verifier feedback, thereby increasing the probability of producing the correct answer as inference progresses. We further validate these results through controlled experiments.

## 2 PRELIMINARIES

**Transformers.** In this work, we consider attention-only Transformers defined as follows.

**Definition 2.1** (Transformer). We define a Transformer model over vocabulary $\mathcal{V}$ as a tuple

$$(\theta, \mathrm{pe}, (\mathbf{K}_h^{(l)}, \mathbf{Q}_h^{(l)}, \mathbf{V}_h^{(l)})_{h \in [H], l \in [L]}, \vartheta, \mathcal{V})$$

where $\theta : \mathcal{V} \to \mathbb{R}^d$ is the tokenizer, $\mathrm{pe} : \mathbb{R}^d \times \mathcal{V}^\omega \to \mathbb{R}^d$ is a position encoder, $\mathbf{K}_h^{(l)}, \mathbf{Q}_h^{(l)}, \mathbf{V}_h^{(l)} \in \mathbb{R}^{d \times d}$ are the key, query, value matrices over $L$ layers and $H$ heads each layer, and $\vartheta$ is the output feature. The computation of a Transformer rolls out as follows:

1. For each $i = 1, \ldots, n$, $X_i^{(1)} = \mathrm{pe}(\theta(v_i); v_1, \ldots, v_i)$.

2. For each $l = 1, \ldots, L-1$, compute each $X_i^{(l+1)}$ for $i = 1, \ldots, n$ by

$$X_i^{(l+1)} = \sum_{h=1}^{H} \sum_{j=1}^{i} \frac{\exp\left(s_h^{(l)}(X_i, X_j)\right)}{Z_h^{(l)}} \cdot \mathbf{V}_h^{(l)} X_j^{(l)}, \tag{1}$$

   where $s_h^{(l)}(\cdot)$ is the attention score defined by $s_h^{(l)}(X_i, X_j) = (\mathbf{Q}_h^{(l)} X_i^{(l)})^\top (\mathbf{K}_h^{(l)} X_j^{(l)})$ and $Z_h^{(l)} = \sum_{j=1}^{i} \exp\left(s_h^{(l)}(X_i, X_j)\right)$ is the normalizing constant.

3. The output probability is given by

$$p_f(y|v_1, \ldots, v_n) = \mathrm{Softmax}(\vartheta(y)^\top X_n^{(L)}), \ y \in \mathcal{V}.$$

In particular, we assume the softmax attention layer has precision $\epsilon$: if two attention scores $s_1, s_2$ satisfy $e^{s_1} < \epsilon \cdot e^{s_2}$, then $e^{s_1}$ is treated as zero in the attention computation of Eq. (1).

While classical positional encoders are solely dependent on the index of the current token (i.e. we may write $\mathrm{pe}(\theta(v_i); v_1, \ldots, v_i) = \mathrm{pe}(\theta(v_i); i)$), recent advances (He et al., 2024; Zhang et al., 2024b; Golovneva et al., 2024) have extended this notion to incorporate set membership information of preceding tokens. This generalization proves crucial for enhancing the long-context capability required for effective self-correction. Motivated by this insight, we introduce the following notion.

**Definition 2.2** (Generalized Position Encoder). We say that $\mathrm{pe} : \mathbb{R}^d \times \mathcal{V}^\omega \to \mathbb{R}^d$ is a generalized position encoder w.r.t. a partition $\mathcal{V}_1, \ldots, \mathcal{V}_K$ of $\mathcal{V}$ if it maps an input feature in $\mathbb{R}^d$ and a token sequence (of arbitrary length) $v_1, \cdots, v_i$ to a vector in $\mathbb{R}^d$, so that it only depends on the input feature and the membership of each $v_i$ in the sets $\mathcal{V}_1, \ldots, \mathcal{V}_K$, i.e.

$$\mathrm{pe}(\theta(v_i); v_1, \ldots, v_i) = \mathrm{pe}\left(\theta(v_i); (\mathbb{1}(v_j \in \mathcal{V}_k))_{j \in [i], k \in [K]}\right).$$

**Test-time scaling.** In this work, we study the following three strategies for test-time scaling.

1. *Self-consistency* samples $n$ i.i.d. responses from the language model and chooses the most consistent answer, while marginalizing over the reasoning paths.

2. *Best-of-n* samples $n$ i.i.d. responses from the language model and chooses the answer with the highest score given by the reward model.

3. In the *Self-Correction* paradigm, the Transformer autonomously generates a sequence of $n$ responses, each conditioned on the previous responses and their respective reward scores.

# 3 SEPARATION BETWEEN SELF-CONSISTENCY AND BEST-OF-N

In this section, we study the sample complexity of self-consistency and best-of-$n$. Let $q$ denote the user query (e.g. a math problem) and $\mathcal{O}$ denote the answer space; then for each answer $o \in \mathcal{O}$ we define $p(o)$ as the marginalized probability of generating $o$ over all possible reasoning paths

$$p(o) = \sum_{\text{reasoning path}} p_f(\text{reasoning path}, o|q)$$

where $p_f$ denotes the probability distribution of Transformer $f$.

To understand the sample complexity, we focus on the dependence on the following probability gap:

$$\Delta := p(o^*) - \max_{o \in \mathcal{O}, o \neq o^*} p(o)$$

where $o^*$ denotes the correct answer[1]. If $\Delta \leq 0$, then self-consistency fails to find the correct answer with high probability and the separation becomes trivial. Therefore, we focus on the setting where $\Delta > 0$ (i.e., the most likely answer is correct), which is also considered in prior theoretical work (Huang et al., 2024a). Under this setting, we assume that the reward function $r$ is maximized (only) at the correct answer, because $p$ itself is such a reward function satisfying this condition. Note that since $p(o)$ is marginalized over reasoning paths, $\Delta > 0$ does not imply that the correct answer can be derived easily from greedy decoding.

**Theorem 3.1** (Sample Complexity of Self-Consistency). *Let $O = |\mathcal{O}| < \infty$. There exist numerical constants $c, C > 0$ such that: when $n \geq \frac{C \log(O/\delta)}{\Delta^2}$, self-consistency with $n$ i.i.d. samples is able to produce the correct answer with probability at least $1 - \delta$; when $n \leq \frac{c}{\Delta^2}$, there exists a hard instance where self-consistency with $n$ i.i.d. samples fails to produce the correct answer with constant probability.*

**Theorem 3.2** (Sample Complexity of Best-of-$n$). *Assume $r(o^*) > r(o), \forall o \neq o^*$. There exist numerical constants $c, C > 0$ such that: when $n \geq \frac{C \log(1/\delta)}{\Delta}$, best-of-$n$ with $n$ samples produces the correct answer with probability at least $1 - \delta$; When $n \leq \frac{c}{\Delta}$, there exists a hard instance where best-of-$n$ with $n$ i.i.d. samples fails to produce the correct answer with constant probability.*

By providing matching (up to logarithmic factors) upper and lower bounds on the number of samples, the above results establish the separation between self-consistency and best-of-$n$. While self-consistency requires $\Theta(1/\Delta^2)$ samples to produce the correct answer, best-of-$n$ has an advantage, requiring only $\Theta(1/\Delta)$ samples. Therefore, this theory corroborates the empirical findings that best-of-$n$ generally leads to better problem solving accuracy on reasoning tasks compared with self-consistency (Sun et al., 2024; Wu et al., 2025a).

# 4 EXPRESSIVENESS UNDER SELF-CORRECTION

A key distinction between self-correction and the repeated sampling strategies discussed in the previous section lies in the dependence structure of the generated responses: unlike repeated sampling, the outputs produced by self-correction are not i.i.d.. Consequently, to analyze the sample efficiency of self-correction, we must first address a fundamental question: can a large language model (LLM), through self-correction, increase the likelihood of generating the correct answer? At its core, this question is one of expressiveness—whether the Transformer architecture's representation capacity is sufficient to support such improvement.

In this section, we take a first step toward analyzing the expressiveness of Transformers under the self-correction paradigm. Unlike prior work that focuses on expressiveness in the context of a single

---

[1]If multiple correct answers exist, the self-consistency result applies after canonicalizing answers into equivalence classes and assigning probability mass to these classes.

task, we study what we call *general-purpose expressiveness*: the ability to solve a broad range of tasks. To this end, we introduce the concept of a General-Purpose Transformer—a construction that maps any collection of task-specific Transformers (experts) into a single unified Transformer.

**Definition 4.1** (General-Purpose Transformer). We say that $\phi$ is a General-Purpose Transformer of type $(t_1, t_2)$ if it maps any set of Transformers with hidden size $d$ and depth $L$ into another 'unified' Transformer with hidden size $t_1 \cdot d + t_2$ and depth $L + O(1)$.

A general-purpose Transformer provides a principled framework for constructing more powerful Transformer architectures by composing simpler, task-specific components. This meta-architecture enables a single model to solve multiple tasks at inference time, representing a significant advancement in our theoretical understanding of the expressive power of modern machine learning systems. Our goal is to investigate the general-purpose expressiveness of self-correction paradigms through the lens of general-purpose Transformers: specifically, how a Transformer can adaptively solve different tasks during inference without prior knowledge of the task identity.

## 4.1 GENERAL-PURPOSE EXPRESSIVENESS

In this section, we present two auxiliary results that serve as building blocks for constructing general-purpose Transformers capable of solving multiple tasks. These results may also be of independent interest beyond expressiveness of self-correction.

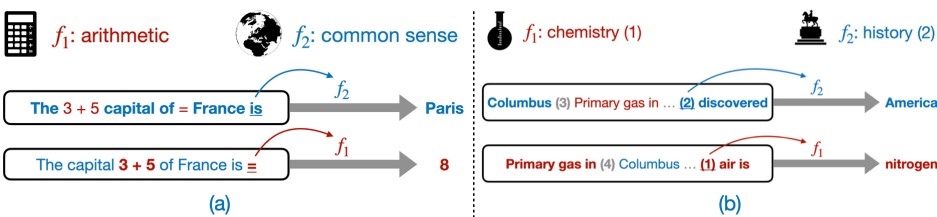

Figure 2: (a): Illustration of Proposition 4.2. In the first query, $f_2$ is called to solve the common sense problem by attending to only blue tokens. In the second query, $f_1$ is called to solve the arithmetic problem by attending to only red tokens. (b): Illustration of Proposition 4.4. In the first query, $f_2$ is called to solve the history problem by attending to only blue tokens. In the second query, $f_1$ is called to solve the chemistry problem by attending to only red tokens. Importantly, these function calls occur implicitly within the internal computation of the unified Transformer architecture.

The first result addresses the setting in which multiple Transformers operate over distinct vocabularies, with each vocabulary corresponding to a specific task. The objective is to construct a unified Transformer that uses the final token in the input sequence to infer which task to perform, and subsequently solves the task by attending only to the task-relevant tokens.

**Proposition 4.2** (General-purpose Expressiveness over Different Token Spaces). *For any $H, L, K, N_{\max} \in \mathbb{Z}_+$, $\mathcal{V}_i \cap \mathcal{V}_j = \emptyset$ ($\forall i \neq j \in \{0\} \cup [K]$), there exists a general-purpose Transformer $\phi$ of type $(O(K), O(\log N_{\max}))$ such that for any Transformers $f_k = (\theta_k, \mathrm{pe}_k, (\mathbf{K}^{(l)}_{k;h}, \mathbf{Q}^{(l)}_{k;h}, \mathbf{V}^{(l)}_{k;h})_{h \in [H], l \in [L]}, \vartheta_k, \mathcal{V}_k)$ for $k \in [K]$, the Transformer $\widetilde{f} = \phi(f_1, \ldots, f_K)$ satisfies the following property: for any token sequence $v = v_1 \cdots v_n$ such that $n \leq N_{\max}$ and there exists one $v_{i_0} \in \mathcal{V}_0$, we have*

$$p_{\widetilde{f}}(\cdot | v) = p_{f_\kappa}(\cdot | u)$$

*where $\kappa$ is the task indicated by the last token: i.e., $v_n \in \mathcal{V}_\kappa$, and $u = v_{i_1} \cdots v_{i_m}$, where $\{i_1 < \cdots < i_m\} = \{i : v_i \in \mathcal{V}_\kappa\}$, is the sequence of tokens relevant to task $\kappa$.*

**Remark 4.3.** *The existence of $v_{i_0}$ which does not belong to any $\{\mathcal{V}_i\}_{i \in [K]}$ serves the technical purpose of inducing attention sink of all irrelevant experts to $v_{i_0}$. It may be achieved by assuming the user query always ends with the special token* `<eos>`.

The following result considers a more challenging scenario in which multiple Transformers operate across different tasks but share a common vocabulary space. A set of indicator tokens, denoted by $\Omega$, is used to specify the intended task. The objective is to determine which task to execute based

on the most recent indicator token. It then proceeds to solve the task by attending exclusively to the task-relevant tokens appearing before the first indicator token and after the last indicator token in the input sequence.

**Proposition 4.4** (Multi-Task Representation over the Same Token Space). *For any $H, L, K, N_{\max} \in \mathbb{Z}_+$, token spaces $\Omega \cap \mathcal{V} = \emptyset$, there exists a general-purpose Transformer $\phi$ of type $(O(K), O(\log N_{\max}))$ such that for any Transformers $f_k = (\theta_k, \mathrm{pe}_k, (\mathbf{K}_{k;h}^{(l)}, \mathbf{Q}_{k;h}^{(l)}, \mathbf{V}_{k;h}^{(l)})_{h \in [H], l \in [L]}, \vartheta_k, \mathcal{V}), k \in [K]$ over $\mathcal{V}$, the Transformer $\widetilde{f} = \phi(f_1, \ldots, f_K)$ satisfies the following property: for any token sequence $v = v_1 \cdots v_n$ such that*

$$\{\xi_1 < \cdots < \xi_m\} = \{j : v_j \in \Omega\}, \, \xi_m < n \le N_{\max}$$

*then we have*

$$p_{\widetilde{f}}(\cdot|v) = p_{f_\kappa}(\cdot|u) \tag{2}$$

*where $u = v_1 \cdots v_{\xi_1 - 1} v_{\xi_m + 1} \cdots v_n$ is the token sequence obtained by omitting tokens from position $\xi_1$ to $\xi_m$, and $\kappa$ is the task indicated by token $v_{\xi_m}$.*

**Remark 4.5.** *We observe that in both results above, reducing the type parameters is generally not feasible. The dependence on $K$ arises from the need to compute features for all $K$ experts corresponding to the user query. Since the model lacks prior knowledge of the task, it must encode all task-relevant information to preserve the ability to invoke any expert at inference time. The $\log(N_{\max})$ scaling stems from the positional encoding: in order to construct $N_{\max}$ nearly orthogonal vectors, the positional embedding must have dimension at least $\log(N_{\max})$.*

### 4.2 GENERAL-PURPOSE EXPRESSIVENESS OF TRANSFORMERS WITH SELF-CORRECTION

In this section we state the main result that establishes general-purpose expressiveness of Transformers with self-correction. We rely on the following notion of regret-minimization Transformer, which expresses the single task of finding the most rewardable action.

**Definition 4.6** (Regret-Minimization Transformer). We say that a Transformer $f$ achieves simple regret $\mathrm{reg}(\cdot)$ over reward function $r$ and action space $\mathcal{A}$, if for any $T \in \mathbb{Z}_+$, we have $\max_{a^* \in \mathcal{A}} r(a^*) - \mathbb{E}[r(a_T)] \le \mathrm{reg}(T) = o(1)$ where $a_1, \ldots, a_T$ are generated in the following way:

$$a_t \sim p_f(\cdot \mid a_1, R_1, \ldots, a_{t-1}, R_{t-1}), \, \forall t = 1, \ldots, T,$$
$$\mathbb{E}[R_t \mid a_t, \mathcal{H}_{t-1}] = r(a_t), \, \forall t = 1, \ldots, T.$$

Essentially, the goal of a regret-minimization Transformer is to learn from a reward oracle and ultimately recommend an action that is near-optimal, which is related to a concept commonly referred to as simple regret in the bandit literature (Even-Dar et al., 2006; Carpentier & Valko, 2015; Jamieson et al., 2014). To achieve this, the Transformer may implement strategies such as mirror descent, upper confidence bounds, or search-based algorithms, depending on the problem structure. As these procedures rely only on basic arithmetic operations, such Transformers can be constructed by applying the universal approximation capabilities of Transformers (Yun et al., 2020; Luo et al., 2022; Feng et al., 2023; Li et al., 2024b): For example, Lin et al. (2023) provide constructions to approximate upper confidence bounds and Thompson sampling algorithms with average regret $O(1/\sqrt{T})$. Consequently, their construction is not the primary focus of this work.

The following theorem establishes the existence of a general-purpose Transformer that can simulate the behavior of a set of expert Transformers (not necessarily over the same token space) through self-correction. Specifically, it shows that such a unified Transformer can, at inference time, identify and invoke the appropriate expert to solve any task that the original experts can solve. The self-correction protocol is described in Algorithm 1, wherein the unified Transformer autoregressively generates actions and responses, after which the verifier is queried to obtain reward signals. Through this process of trial and error, the model effectively "learns" at inference time, using the verifier to minimize regret and adaptively select the correct expert.

**Theorem 4.7** (Regret Minimization via Self-Correction). *For any $H, L, K, N_{\max} \in \mathbb{Z}_+$, token spaces $\mathcal{V}_0, \mathcal{V}_1, \ldots, \mathcal{V}_K, \mathcal{A}$ ($|\mathcal{A}| = K$) such that $\mathcal{V}_0, \mathcal{V} = (\cup_{k=1}^K \mathcal{V}_k)$, and $\mathcal{A}$ are disjoint, and reward function $r$, there exists a general-purpose Transformer $\phi$ of type $(O(K), O(\log N_{\max}))$ such that given any set of Transformers denoted as follows,*

---

**Algorithm 1** Self-correction with verifier

---

1: **procedure** GENERATION($q$)        ▷ $q = q_1 \ldots q_{n_0}$ denotes the user query.
2:     prompt $\leftarrow q$
3:     **for** $t = 1, \ldots, T$ **do**
4:        $a^{(t)} \sim p_{\widetilde{f}}(\cdot \mid \text{prompt})$      ▷ $a^{(t)}$ designates which expert to use in $t$-th iteration
5:        prompt $\leftarrow$ prompt$|a^{(t)}$    ▷ Update the prompt autoregressively, | represents token concatenation.
6:        **for** $i = 1, \ldots$ **do**
7:           $u_i^{(t)} \sim p_{\widetilde{f}}(\cdot \mid \text{prompt})$       ▷ Generate $t$-th response autoregressively
8:           prompt $\leftarrow$ prompt$|u_i^{(t)}$       ▷ Update the prompt autoregressively
9:           **if** $u_i^{(t)} = \text{EOS}$ **then**
10:             **Break**
11:        $r^{(t)} \leftarrow r(q, u^{(t)})$, prompt $\leftarrow$ prompt$|r^{(t)}$    ▷ Query verifier to obtain reward of $t$-th response
12:     **Return** $u^{(T)}$

---

- *K **expert Transformers**:* $f_k = (\theta_k, \text{pe}_k, (\mathbf{K}_{k;h}^{(l)}, \mathbf{Q}_{k;h}^{(l)}, \mathbf{V}_{k;h}^{(l)})_{h \in [H], l \in [L]}, \vartheta_k, \mathcal{V}_k)$ *for* $k \in [K]$, *such that one of the expert* $f_{k^*}$ *achieves* $\lambda$-*suboptimal reward:*

$$\mathbb{E}_{u \sim f_{k^*}(\cdot|q)}[r(q, u)] \geq \max_{u^* \in \mathcal{V}^\omega} r(q, u^*) - \lambda$$

- ***Regret-Minimization Transformer***: $f_0 = (\theta, \text{pe}, \mathbf{K}_{0;h}^{(l)}, \mathbf{Q}_{0;h}^{(l)}, \mathbf{V}_{0;h}^{(l)})_{h \in [H], l \in [L]}, \vartheta, \mathcal{V}_0 \cup \mathcal{A})$ *that implements a bandit algorithm over the reward function* $r_0$ *and action space* $\mathcal{A}$ *with simple regret* $\text{reg}(t)$, *where* $r_0(a) = \mathbb{E}_{u \sim f_a(\cdot|q)}[r(q, u)]$ *denotes the average reward of responses generated by the* $a$-*th expert,*

*then the Transformer* $\widetilde{f} = \phi(f_0, f_1, \ldots, f_K)$ *satisfies the following property: for any prompt* $v = v_1 \cdots v_n$, *if the response sequence* $u^{(1)}, \ldots, u^{(T)}$ *generated by the protocol in Algorithm 1 has total length* $\leq N_{\max}$, *then we have*

$$\max_{u^* \in \mathcal{V}^\omega} r(q, u^*) - \mathbb{E}[r(q, u^{(T)})] \leq \lambda + \text{reg}(T)$$

**Remark 4.8.** *While the general-purpose Transformer* $\phi$ *can be applied to construct the brute-force Transformer* $\widetilde{f}$ *that simply tries every expert, we note that the generality of Definition 4.6 allows us to construct more powerful Transformers beyond brute-force search. Leveraging the structures in the problem and the expert pool, it is entirely possible to identify the correct expert using* $\ll K$ *trials (Russo & Van Roy, 2018; Foster et al., 2021).*

As a consequence of Theorem 4.7, the transformer $\widetilde{f}$ can solve $K$ distinct tasks at inference time, without requiring prior knowledge of the task of identification of the best expert. The architecture-level composition is *general-purpose* in the sense that it does not depend on the internal parameters of the expert Transformers. The regret guarantee, however, depends on the assumed verifier feedback model and on the existence of a regret-minimization Transformer for the induced reward function. To our knowledge, this is among the first theoretical expressiveness analyses of Transformer self-correction in this multi-expert setting. Furthermore, our theory aligns with the empirical finding that LLMs can progressively optimize outcome rewards during test-time (Qu et al., 2025; Song et al., 2025; Tooliense, 2025; Monea et al., 2024).

## 5 EXPERIMENTS

In this section, we conduct synthetic experiments to show that Transformers can self-correct with verifier feedback.[2]

### 5.1 EXPRESSIVENESS OF SELF-CORRECTION

**Data generation.** We aim to construct a test problem with complex prompts such that correctly solving the problem in the single-turn generation is challenging. In this case, self-correction can

---

[2]Code: `https://github.com/LithiumDA/Representation-Ability-of-Test-time-Scaling`.

play a critical role if Transformers have this capability. Specifically, in our synthetic problem, the prompt is the concatenation of the following two components:

- **Instruction**: A 3-SAT problem, e.g.,

$$(\sim x_3 \vee \sim x_1 \vee \sim x_2) \wedge (\sim x_1 \vee \sim x_3 \vee x_2) \wedge (\sim x_4 \vee x_2 \vee \sim x_3) \wedge \cdots$$

- **Data**: A string composed of characters from the set $\{\texttt{a}, \texttt{b}\}$.

The ground truth target is defined as follows: If the 3-SAT problem in the *instruction* is satisfiable, the model should *copy* the string in the *data* part in the output; otherwise, the model should *reverse* the string in the output. In our experiment, we construct datasets using 3-SAT problems with 4 variables and 20 clauses. The lengths of the data strings are set to 5. We generate 10000 instances for training and 512 instances for evaluation. In the training set, we control the ratio of satisfiable and unsatisfiable 3-SAT instructions to 9:1, while in the test set, the ratio is set to 1:1. This label imbalance encourages models to fail on their first attempt, thereby eliciting self-correction behavior.

**Models and training configuration.**  We train a class of Transformer models of various sizes: {GPT-nano, GPT-micro, GPT-mini, Gopher-44M} with the Adam optimizer (Kingma & Ba, 2015) for 5 epochs. More implementation details can be found in Appendix B.

**Results.**  Test set accuracy across different inference settings is shown in Figure 3. We note that model performance plateaus at $63.19\%$ when there is no self-correction at test time, with no improvement from increased model size. By contrast, when models are equipped with verifier signals to enable self-correction, test accuracy improves substantially, demonstrating the efficacy of this mechanism. Crucially, larger models – such as GPT-mini and Gopher-44M – achieve near-perfect accuracy under self-correction, suggesting that sufficiently expressive Transformers are capable of implementing effective self-correction strategies. This empirical result supports our theoretical findings.

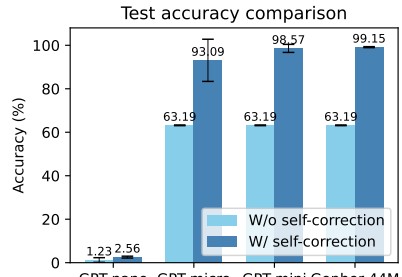

Figure 3: Accuracy comparisons of different models with/without self-correction at test time.

## 5.2 EVALUATION OF SAMPLE COMPLEXITY

**Dataset.**  We conduct experiments on the AIME 2024 & 2025 datasets (Mathematical Association of America, 2025), which serve as a real-world benchmark for evaluating mathematical reasoning tasks. These datasets allow us to measure not only the raw accuracy of different large language models (LLMs), but also the impact of verification-based strategies on sample efficiency.

**Model configuration.**  We consider recent LLMs, including `Qwen3-1.7B`, `Qwen3-4B` (Yang et al., 2025), and `Llama-3.2-3B-Instruct` (Dubey et al., 2024), as candidate models. In addition, `Qwen3-32B` is employed as an LLM verifier. This setup enables us to compare standard decoding strategies (self-consistency) with verification-based methods (best-of-$n$ and self-correction).

**Results.**  We compare the accuracy of self-consistency, best-of-$n$, and self-correction under different sample sizes. Notably, as summarized in Table 1, best-of-$n$ with only 4 samples consistently outperforms self-consistency with 64 samples, confirming the predicted gap in sample complexity.

| Model \ Method | Self-consistency (64 samples) | Best-of-$n$ (4 samples) | Self-correction (4 samples) |
|---|---|---|---|
| Qwen3-1.7B | 58.33% | 59.68% | 79.29% |
| Qwen3-4B | 78.33% | 80.58% | 81.19% |
| Llama-3.2-3B-Instruct | 1.67% | 4.84% | 24.52% |

Table 1: Accuracy comparison of self-consistency, best-of-$n$, and self-correction methods on AIME 24 & 25 datasets.

Furthermore, self-correction with verifiers achieves strong performance, highlighting the ability of LLMs to leverage verifier feedback effectively. These results show a notable sample complexity gap between Self-consistency and Best-of-$n$ and demonstrate that modern Transformer models are sufficiently expressive to implement self-correction mechanisms when combined with verifiers, thus consistent with our theoretical results in Section 3 and 4.

## 6 RELATED WORK

**Theories of Transformers and Large Language Models.** The success of Transformers and LLMs has motivated the study on their expressiveness. Existing research has shown that Transformers can implement simple functions such as sparse linear functions, two-layer neural networks, and decision trees (Garg et al., 2022), gradient descent (Akyürek et al., 2022; Bai et al., 2023; Von Oswald et al., 2023), automata (Liu et al., 2022; Zhao et al., 2023), Dyck languages (Bhattamishra et al., 2020a; Yao et al., 2021), Turing machines (Dehghani et al., 2018; Bhattamishra et al., 2020b; Zaheer et al., 2020; Pérez et al., 2021; Wei et al., 2022a), variational inference (Mei & Wu, 2023), and bandit algorithms (Lin et al., 2023). Yun et al. (2020); Luo et al. (2022); Alberti et al. (2023); Petrov et al. (2024) establish universal approximation results under various settings. Edelman et al. (2022); Elhage et al. (2021); Li et al. (2021); Likhosherstov et al. (2021) study representational capabilities and properties of self-attention, the core component in Transformers. Feng et al. (2023); Li et al. (2024b) study the expressiveness of auto-regressive Transformers with chain-of-thought. Edelman et al. (2022); Li et al. (2024a); Botta et al. (2025) study the sample complexity of Transformers. Recently, a growing body of work has begun to explore the theoretical foundations of self-improvement in large language models (LLMs). Song et al. (2024b) introduce the generation-verification gap as a key quantity governing scaling behavior. Huang et al. (2024a) propose a progressive sharpening framework in which the policy gradually shifts toward more confident responses. Setlur et al. (2025) draw on reinforcement learning theory to formally establish the advantages of verifier-based methods. In contrast to these works, our results provide explicit sample complexity rates and tangible representation architectures, enabling a more concrete understanding of the fundamental capabilities and limitations of test-time scaling paradigms.

**Test-time scaling.** Recent research has established the test-time scaling law of LLMs, illuminating a new scaling axis beyond training-time scaling laws (Kaplan et al., 2020; Hoffmann et al., 2022). Existing approaches of scaling up test-time compute of LLMs can be broadly classified into two categories: (1) applying test-time algorithms (aka inference-time algorithms) during LLM decoding (Brown et al., 2024; Wu et al., 2025a; Snell et al., 2025); and (2) explicitly training LLMs to output long chain-of-thought traces (DeepSeek-AI, 2025; Team, 2025; OpenAI, 2025; Yang et al., 2025). Many recent works focus on understanding and improving the effectiveness of test-time scaling empirically: Chen et al. (2024c); Aggarwal & Welleck (2025); Cuadron et al. (2025); Wang et al. (2025) study under-thinking, over-thinking, and length control in LLM reasoning. Chen et al. (2025) proposes to integrate self-verification and self-correction into sampling. Qu et al. (2025) analyze optimizing test-time compute by introducing a meta reinforcement learning formulation. Setlur et al. (2025) demonstrate that verification/RL is important for optimal test-time scaling. Zhang et al. (2025) provide an extensive review of the test-time scaling landscape. In contrast, our work focuses on theoretical analyses of test-time scaling. In addition, our work provides theoretical explanation of In-Context Reinforcement Learning (Song et al., 2025; Tooliense, 2025; Monea et al., 2024).

## 7 DISCUSSIONS

Our investigation reveals a fundamental separation in sample complexity between self-consistency and best-of-$n$, providing theoretical support for the empirically observed superiority of the latter method. Furthermore, by introducing the framework of *general-purpose expressiveness*, we construct generic Transformer architectures capable of emulating online learning algorithms at test time. This capability enables a single model to provably solve multiple tasks without task-specific adaptation, thus extending our understanding of expressiveness to multi-task settings. Our experiments validate the theoretical separation and confirm that it requires additional model capacities for Transformer to implement self-correction.

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

# A PROOFS

## A.1 PROOF OF THEOREM 3.1

*Proof.* **Upper bound.** Let $N(o)$ denote the number of samples equal to $o$. For each $o \neq o^\star$, define

$$Z_t^{(o)} := \mathbf{1}\{X_t = o^\star\} - \mathbf{1}\{X_t = o\} \in [-1, 1].$$

Then

$$\mathbb{E}[Z_t^{(o)}] = p(o^\star) - p(o) \geq \Delta.$$

By Hoeffding's inequality,

$$\mathbb{P}(N(o^\star) \leq N(o)) = \mathbb{P}\left(\sum_{t=1}^n Z_t^{(o)} \leq 0\right) \leq \exp\left(-\frac{n\Delta^2}{2}\right).$$

Taking a union bound over all $O - 1$ incorrect answers gives

$$\mathbb{P}(\exists o \neq o^\star : N(o) \geq N(o^\star)) \leq (O - 1)\exp\left(-\frac{n\Delta^2}{2}\right) \leq \delta.$$

**Lower bound.** Consider the two-answer instance

$$p(o^\star) = \frac{1 + \Delta}{2}, \qquad p(o_2) = \frac{1 - \Delta}{2}.$$

Self-consistency succeeds iff $N(o^\star) > N(o_2)$ up to tie-breaking. Let

$$S_n := \sum_{t=1}^n Z_t, \qquad Z_t = \begin{cases} +1, & X_t = o^\star, \\ -1, & X_t = o_2. \end{cases}$$

Then $\mathbb{E}Z_t = \Delta$ and $\mathrm{Var}(Z_t) = 1 - \Delta^2$. For $n \leq c/\Delta^2$, choosing a sufficiently small numerical constant $c > 0$, Claim A.5 gives

$$\Pr(S_n \leq 0) \geq c_0$$

for a universal constant $c_0 > 0$. Therefore self-consistency fails with constant probability.

$\square$

## A.2 PROOF OF THEOREM 3.2

*Proof.* Write $\mathcal{O} = \{1, \ldots, O\}$ where $i$ is the $i$-th most likely answer and let $n_i$ denote the number of occurrences of $i$. Then we have

$$p(1) \geq p(2) + \Delta \geq \Delta.$$

Note that for best-of-$n$, correctness is achieved if the correct answer appears at least once among $n$ independent samples.

**Upper bound.** When $n \geq \frac{2\log(1/\delta)}{\Delta}$, we have

$$\mathbb{P}(\text{Best-of-}n \text{ outputs correct answer}) = 1 - (1 - p(1))^n$$
$$\geq 1 - (1 - \Delta)^{\frac{2\log(1/\delta)}{\Delta}}$$
$$\geq 1 - \delta.$$

This confirms that best-of-$n$ achieves the correct answer with $1 - \delta$ probability.

**Lower bound.** Choose $O \geq (1 - \Delta)/\Delta$ and set

$$p(1) = \Delta + \frac{1 - \Delta}{O}, \qquad p(2) = \cdots = p(O) = \frac{1 - \Delta}{O}.$$

Then $p(1) - p(2) = \Delta$ and $p(1) \leq 2\Delta$. Hence, for $n \leq 1/\Delta$,

$$\Pr(\text{success}) = 1 - (1 - p(1))^n \leq 1 - (1 - 2\Delta)^{1/\Delta} \leq 1 - e^{-4} < 0.99$$

for sufficiently small $\Delta$.

This confirms that best-of-$n$ fails to produce the correct answer with constant probability. $\square$

## A.3 PROOF OF PROPOSITION 4.2

We first introduce the following result that extends any Transformer to a larger vocabulary, so that it only attends to tokens in its original vocabulary.

**Proposition A.1** (Extended Representation to Multiple Token Spaces). *For any $H, L, N_{\max} \in \mathbb{Z}_+$, $\mathcal{V}_1 \cap \mathcal{V}_0 = \emptyset$, there exists a general-purpose Transformer $\phi$ of type $(O(1), O(\log N_{\max}))$ such that for any Transformers $f = (\theta, \mathrm{pe}, (\mathbf{K}_h^{(l)}, \mathbf{Q}_h^{(l)}, \mathbf{V}_h^{(l)})_{h \in [H], l \in [L]}, \vartheta, \mathcal{V}_1)$ over vocabulary $\mathcal{V}_1$, the Transformer $\widetilde{f} = \phi(f)$ satisfies the following property: for any token sequence $v = v_1 \cdots v_n$ such that $n \leq N_{\max}$, denote $\{i_1 < \cdots < i_m\} = \{i : v_i \in \mathcal{V}_1\}$, then we have*

$$p_{\widetilde{f}}(\cdot|v) = p_f(\cdot|u),$$

*where $u = v_{i_1} \cdots v_{i_m}$.*

*Proof.* Set constants $B_v, B_{qk}, B_\theta$ such that for any layer $l$ and head $h$, it holds that $\left\|(\mathbf{Q}_h^{(l)})^\top \mathbf{K}_h^{(l)}\right\|_2 \leq B_{qk}$, $\left\|\mathbf{V}_h^{(l)}\right\|_2 \leq B_v$, and $\|\theta(v)\|_2 \leq B_\theta$ holds for all $v \in \mathcal{V}$. Let $B = (HB_v)^L B_{qk} B_\theta, C = 4B^2 + \log(1/\epsilon), C_0 = 4C$. By Lemma A.3, there exists $\alpha_1, \ldots, \alpha_{N_{\max}}, \beta_0, \beta_1 \in \mathbb{R}^{d_0}$ and $A_0, A_1, A \in \mathbb{R}^{d_0 \times d_0}$ for $d_0 \leq O(\log N_{\max})$ such that

1. For any $i \geq j_1, j_2, j_3$:
$$(\alpha_i + \beta_1)^\top A_0 (\alpha_{j_1} + \beta_1) = (\alpha_i + \beta_1)^\top A_0 (\alpha_{j_2} + \beta_1) \geq (\alpha_i + \beta_1)^\top A_0 (\alpha_{j_1} + \beta_0) + C_0$$
$$(\alpha_i + \beta_0)^\top A_0 (\alpha_i + \beta_0) \geq (\alpha_i + \beta_0)^\top A_0 (\alpha_{j_1} + \beta_1) + C_0, \tag{3}$$

2. For any $i > j$
$$(\alpha_i + \beta_1)^\top A (\alpha_i + \beta_1) \geq (\alpha_i + \beta_1)^\top A (\alpha_j + \beta_1) + C_0$$
$$\geq (\alpha_i + \beta_1)^\top A (\alpha_j + \beta_0) + 2C_0, \tag{4}$$

3. For any $i \geq j, j_1$
$$(\alpha_i + \beta_1)^\top A_1 (\alpha_j + \beta_0) = (\alpha_i + \beta_1)^\top A_1 (\alpha_{j_1} + \beta_1) + C_0$$
$$(\alpha_i + \beta_1)^\top A_1 (\alpha_i + \beta_1) \geq \max\{(\alpha_i + \beta_1)^\top A_1 (\alpha_{j_1} + \beta_1), (\alpha_i + \beta_1)^\top A_1 (\alpha_{j_1} + \beta_0)\} + C_0. \tag{5}$$

We define $\phi$ as follows: for any Transformers $f = (\theta, \mathrm{pe}, (\mathbf{K}_h^{(l)}, \mathbf{Q}_h^{(l)}, \mathbf{V}_h^{(l)})_{h \in [H], l \in [L]}, \vartheta, \mathcal{V}_1)$, the Transformer $\widetilde{f} = \phi(f)$ is given by

$$(\widetilde{\theta}, \widetilde{\mathrm{pe}}, (\widetilde{\mathbf{K}}_h^{(l)}, \widetilde{\mathbf{Q}}_h^{(l)}, \widetilde{\mathbf{V}}_h^{(l)})_{h \in [H+1], l \in [L]}, \widetilde{\vartheta}, \mathcal{V}_1 \cup \mathcal{V}_0),$$

where the tokenizer is given by

$$\widetilde{\theta}(v) = \mathbb{1}(v \in \mathcal{V}_1) \cdot \begin{pmatrix} \theta(v) \\ \beta_1 \end{pmatrix} + \mathbb{1}(v \in \mathcal{V}_0) \cdot \begin{pmatrix} 0 \\ \beta_0 \end{pmatrix},$$

the positional encoder is given by

$$\widetilde{\mathrm{pe}}\left(\begin{pmatrix} x \\ y \end{pmatrix}; v_1, \ldots, v_i\right) = \begin{pmatrix} \mathrm{pe}(x; u) \\ \alpha_i + y \end{pmatrix},$$

where $u = v_{i_1} \cdots v_{i_m}$ and $x \in \mathbb{R}^d$; for $l = 1, \ldots, L$ the key, query, value matrices are given by

$$\widetilde{\mathbf{K}}_h^{(l)} = \begin{pmatrix} \mathbf{K}_h^{(l)} & \\ & A_0 \end{pmatrix}, \quad \widetilde{\mathbf{Q}}_h^{(l)} = \begin{pmatrix} \mathbf{Q}_h^{(l)} & \\ & I \end{pmatrix},$$

$$\widetilde{\mathbf{V}}_h^{(l)} = \begin{pmatrix} \mathbf{V}_h^{(l)} & \\ & 0 \end{pmatrix},$$

$$\widetilde{\mathbf{K}}_{H+1}^{(l)} = \begin{pmatrix} 0 & \\ & A \end{pmatrix}, \quad \widetilde{\mathbf{Q}}_{H+1}^{(l)} = \begin{pmatrix} 0 & \\ & I \end{pmatrix}, \quad \widetilde{\mathbf{V}}_{H+1}^{(l)} = \begin{pmatrix} 0 & \\ & I \end{pmatrix}.$$

The output feature is given by $\widetilde{\vartheta}(y) = \begin{pmatrix} \vartheta(y) \\ 0 \end{pmatrix}$. Since $i_1, \ldots, i_m$ only depends on whether $v_i$'s belong to the set $\mathcal{V}_1$, the generalized position encoding pe is well-defined. It can be verified that $\phi$ is indeed a general-purpose Transformer of type $(O(1), O(\log N_{\max}))$.

We show that for any $l = 1, \ldots, L$,

$$\widetilde{X}_i^{(l)} = \begin{pmatrix} X_i^{(l)} \\ \widetilde{\alpha}_i \end{pmatrix}, \ \forall i = i_1, \ldots, i_m \tag{6}$$

where $X_i^{(l)}$ is the $l$-th layer of Transformer $f$ at position $i$ (attending only to positions $i_1, \ldots, i_m$) such that

$$\|X_i^{(l)}\|_2 \le B_\theta (HB_v)^l, \tag{7}$$

and

$$\widetilde{X}_j^{(l)} = \begin{pmatrix} 0 \\ \widetilde{\alpha}_j \end{pmatrix}, \ \forall j \notin \{i_1, \ldots, i_m\} \tag{8}$$

where $\widetilde{\alpha}_i = \alpha_i + \mathbb{1}(v \in \mathcal{V}_0) \cdot \beta_0 + \mathbb{1}(v \in \mathcal{V}_1) \cdot \beta_1$.

We prove these results by induction. The case $l = 1$ folows directly from the definitions of the tokenizer.

**Prove Eq. (6).** Suppose Eq. (6) and Eq. (8) hold for $1, \ldots, l-1$-th layer, and consider $l$-the layer. We have

$$\widetilde{X}_i^{(l+1)} = \underbrace{\sum_{h=1}^{H} \sum_{j=1}^{i} \frac{\exp\left( (\widetilde{\mathbf{Q}}_h^{(l)} \widetilde{X}_i^{(l)})^\top (\widetilde{\mathbf{K}}_h^{(l)} \widetilde{X}_j^{(l)}) \right)}{\widetilde{Z}_h^{(l)}} \cdot \widetilde{\mathbf{V}}_h^{(l)} \widetilde{X}_j^{(l)}}_{\text{term 1}}$$

$$+ \underbrace{\sum_{j=1}^{i} \frac{\exp\left( (\widetilde{\mathbf{Q}}_{H+1}^{(l)} \widetilde{X}_i^{(l)})^\top (\widetilde{\mathbf{K}}_{H+1}^{(l)} \widetilde{X}_j^{(l)}) \right)}{\widetilde{Z}_{H+1}^{(l)}} \cdot \widetilde{\mathbf{V}}_{H+1}^{(l)} \widetilde{X}_j^{(l)}}_{\text{term 2}}.$$

Eq. (3) ensures that for any $i, i' \in \{i_1, \ldots, i_m\}, j \notin \{i_1, \ldots, i_m\}$:

$$(\widetilde{\mathbf{Q}}_h^{(l)} \widetilde{X}_i^{(l)})^\top (\widetilde{\mathbf{K}}_h^{(l)} \widetilde{X}_{i'}^{(l)}) = (\mathbf{Q}_h^{(l)} \widetilde{X}_i^{(l)})^\top (\mathbf{K}_h^{(l)} \widetilde{X}_{i'}^{(l)}) + (\alpha_i + \beta_1)^\top A_0 (\alpha_{i'} + \beta_1)$$

$$\ge (\mathbf{Q}_h^{(l)} X_i^{(l)})^\top (\mathbf{K}_h^{(l)} X_j^{(l)}) + (\alpha_i + \beta_1)^\top A_0 (\alpha_j + \beta_0) + C$$

$$= (\widetilde{\mathbf{Q}}_h^{(l)} \widetilde{X}_i^{(l)})^\top (\widetilde{\mathbf{K}}_h^{(l)} \widetilde{X}_j^{(l)}) + C,$$

and if $i, j_1, j_2 \in \{i_1, \ldots, i_m\}$

$$(\widetilde{\mathbf{Q}}_h^{(l)} \widetilde{X}_i^{(l)})^\top (\widetilde{\mathbf{K}}_h^{(l)} \widetilde{X}_{j_1}^{(l)}) - (\widetilde{\mathbf{Q}}_h^{(l)} \widetilde{X}_i^{(l)})^\top (\widetilde{\mathbf{K}}_h^{(l)} \widetilde{X}_{j_2}^{(l)})$$

$$= (\mathbf{Q}_h^{(l)} X_i^{(l)})^\top (\mathbf{K}_h^{(l)} X_{j_1}^{(l)}) + (\alpha_i + \beta_1)^\top A_0 (\alpha_{j_1} + \beta_1) - (\mathbf{Q}_h^{(l)} X_i^{(l)})^\top (\mathbf{K}_h^{(l)} X_{j_2}^{(l)}) - (\alpha_i + \beta_1)^\top A_0 (\alpha_{j_2} + \beta_1)$$

$$= (\mathbf{Q}_h^{(l)} X_i^{(l)})^\top (\mathbf{K}_h^{(l)} \widetilde{X}_{j_1}^{(l)}) - (\mathbf{Q}_h^{(l)} \widetilde{X}_i^{(l)})^\top (\mathbf{K}_h^{(l)} X_{j_2}^{(l)}),$$

where we use the fact that $C_0 \ge C + 2 \max_{h,l,i,j} \left| (\mathbf{Q}_h^{(l)} X_i^{(l)})^\top (\mathbf{K}_h^{(l)} X_j^{(l)}) \right|$. Since the transformers have precision $\epsilon$ and $C \ge 2 \max_{h,l,i,j} \left| (\mathbf{Q}_h^{(l)} X_i^{(l)})^\top (\mathbf{K}_h^{(l)} X_j^{(l)}) \right| + \log(1/\epsilon)$, it follows that the attention weights of head $(k-1)H + h$ is identical to the attention weights of expert $k$, i.e.

$$\frac{\exp\left( (\widetilde{\mathbf{Q}}_h^{(l)} \widetilde{X}_i^{(l)})^\top (\widetilde{\mathbf{K}}_h^{(l)} \widetilde{X}_j^{(l)}) \right)}{\widetilde{Z}_h^{(l)}} = \mathbb{1}(j \in \{i_1, \ldots, i_m\}) \cdot \frac{\exp\left( (\mathbf{Q}_h^{(l)} X_i^{(l)})^\top (\mathbf{K}_h^{(l)} X_j^{(l)}) \right)}{Z_h^{(l)}}.$$

Therefore

$$\text{term 1} = \sum_{h=1}^{H} \sum_{j=i_1, \ldots, i_m} \frac{\exp\left( (\mathbf{Q}_h^{(l)} X_i^{(l)})^\top (\mathbf{K}_h^{(l)} X_j^{(l)}) \right)}{Z_h^{(l)}} \cdot \begin{pmatrix} \mathbf{V}_h^{(l)} X_j^{(l)} \\ 0 \end{pmatrix} = \begin{pmatrix} X_j^{(l+1)} \\ 0 \end{pmatrix}.$$

Furthermore, by Eq. (4) we have for any $j < i$

$$
\begin{aligned}
(\widetilde{\mathbf{Q}}_{H+1}^{(l)} \widetilde{X}_i^{(l)})^\top (\widetilde{\mathbf{K}}_{H+1}^{(l)} \widetilde{X}_i^{(l)}) &= \widetilde{\alpha}_i^\top A \widetilde{\alpha}_i \\
&\geq \widetilde{\alpha}_i^\top A \widetilde{\alpha}_j + C \\
&= (\widetilde{\mathbf{Q}}_{H+1}^{(l)} \widetilde{X}_i^{(l)})^\top (\widetilde{\mathbf{K}}_{H+1}^{(l)} \widetilde{X}_j^{(l)}) + C,
\end{aligned}
$$

and hence the attention weights concentrate on $i$ itself. Thus

$$
\text{term 2} = \begin{pmatrix} 0 \\ & I \end{pmatrix} \cdot \begin{pmatrix} X_i^{(l)} \\ \widetilde{\alpha}_i \end{pmatrix} = \begin{pmatrix} 0 \\ \widetilde{\alpha}_i \end{pmatrix}.
$$

Combining, we derive Eq.(6) for $(l + 1)$-th layer.

**Prove Eq. (7).** From above,

$$
\begin{aligned}
\|X_i^{(l+1)}\|_2 &= \left\| \sum_{h=1}^{H} \sum_{j=1}^{i} \frac{\exp\left( (\widetilde{\mathbf{Q}}_h^{(l)} \widetilde{X}_i^{(l)})^\top (\widetilde{\mathbf{K}}_h^{(l)} \widetilde{X}_j^{(l)}) \right)}{\widetilde{Z}_h^{(l)}} \cdot \mathbf{V}_h^{(l)} X_j^{(l)} \right\|_2 \\
&\leq H B_v \cdot \max_{j \leq i} \|X_j^{(l)}\|_2 \\
&\leq B_\theta (H B_v)^{l+1}.
\end{aligned}
$$

This confirms Eq. (24) for $l + 1$.

**Prove Eq. (8).** Notice that Eq. (3) ensures that for any $j, j' \notin \{i : v_i \in \mathcal{V}_1\}$ and $i \in \{i : v_i \in \mathcal{V}_1\}$:

$$
\begin{aligned}
(\widetilde{\mathbf{Q}}_h^{(l)} \widetilde{X}_j^{(l)})^\top (\widetilde{\mathbf{K}}_h^{(l)} \widetilde{X}_{j'}^{(l)}) &= (\mathbf{Q}_h^{(l)} X_j^{(l)})^\top (\mathbf{K}_h^{(l)} X_{j'}^{(l)}) + (\alpha_j + \beta_0)^\top A_0 (\alpha_{j'} + \beta_0) \\
&\geq (\mathbf{Q}_h^{(l)} X_j^{(l)})^\top (\mathbf{K}_h^{(l)} X_i^{(l)}) + (\alpha_j + \beta_0)^\top A_0 (\alpha_i + \beta_1) + C \\
&= (\widetilde{\mathbf{Q}}_h^{(l)} \widetilde{X}_j^{(l)})^\top (\widetilde{\mathbf{K}}_h^{(l)} \widetilde{X}_i^{(l)}) + C.
\end{aligned}
$$

It follows that the attention weights is concentrated on the complement of $\{i : v_i \in \mathcal{V}_1\}$ itself, and therefore Eq. (8) follows by a simple induction argument.

Finally, at the output layer

$$
\begin{aligned}
p_{\widetilde{f}}(y|v_1, \ldots, v_n) &= \text{Softmax}(\widetilde{\vartheta}(y)^\top \widetilde{X}_n^{(L)}) \\
&= \text{Softmax}(\vartheta(y)^\top X_m^{(L)}) \\
&= p_f(y|u).
\end{aligned}
$$

This establishes the desired statement. $\qquad \square$

Now we return to the proof of Proposition 4.2.

*Proof.* By Proposition A.1, it suffices to construct general-purpose Transformer $\phi$ such that

$$
p_{\widetilde{f}}(\cdot|v) = p_{f_\kappa}(\cdot|u),
$$

where $u = v_1 \cdots v_{i_0-1} v_{i_0+1} \cdots v_n$, because then the $\widetilde{\phi}$ given by

$$
\widetilde{\phi}(f_1, \ldots, f_K) = \phi(\phi_e(f_1), \ldots, \phi_e(f_K))
$$

satisfies the requirement, where $\phi_e$ is the general-purpose Transformer that extends the $K$ Transformers to the larger vocabulary $\mathcal{V} := \cup_{k=1}^{K} \mathcal{V}_k$ as given by Proposition A.1.

Set constants $B_v, B_{qk}, B_\theta$ such that for any layer $l$ and head $h$, it holds that $\left\| (\mathbf{Q}_h^{(l)})^\top \mathbf{K}_h^{(l)} \right\|_2 \leq B_{qk}$, $\left\| \mathbf{V}_h^{(l)} \right\|_2 \leq B_v$, and $\|\theta(v)\|_2 \leq B_\theta$ holds for all $v \in \mathcal{V}$. Let $B = (KHB_v)^L B_{qk} B_\theta, C = 4B^2 + \log(1/\epsilon), C_0 = 4C$. By Lemma A.3, there exists $\alpha_1, \ldots, \alpha_N, \beta_0, \beta_1, \ldots, \beta_K \in \mathbb{R}^{d_0}$ and $A, A_0, A_1, \ldots, A_K \in \mathbb{R}^{d_0 \times d_0}$ for $d_0 \leq O(K + \log N_{\max})$ such that

1. For any $i \geq j_1, j_2, j_3$ and $k, k', k'' \neq 0$:

$$(\alpha_i + \beta_k)^\top A_0(\alpha_{j_1} + \beta_{k'}) = (\alpha_i + \beta_k)^\top A_0(\alpha_{j_2} + \beta_{k''}) \geq (\alpha_i + \beta_k)^\top A_0(\alpha_{j_1} + \beta_0) + C_0$$

$$(\alpha_i + \beta_0)^\top A_0(\alpha_i + \beta_0) \geq (\alpha_i + \beta_0)^\top A_0(\alpha_{j_1} + \beta_k) + C_0, \tag{9}$$

2. For any $i > j$ and $k \neq k' \neq 0$

$$(\alpha_i + \beta_k)^\top A(\alpha_i + \beta_k) \geq (\alpha_i + \beta_k)^\top A(\alpha_j + \beta_{k'}) + C_0$$
$$\geq (\alpha_i + \beta_k)^\top A(\alpha_j + \beta_0) + 2C_0, \tag{10}$$

3. For any $i \geq j, j_1$ and $k \neq k', k''$

$$(\alpha_i + \beta_k)^\top A_{k'}(\alpha_j + \beta_0) \geq (\alpha_i + \beta_k)^\top A_{k'}(\alpha_{j_1} + \beta_{k''}) + C_0$$

$$(\alpha_i + \beta_k)^\top A_k(\alpha_i + \beta_k) \geq \max\{(\alpha_i + \beta_k)^\top A_k(\alpha_{j_1} + \beta_{k''}), (\alpha_i + \beta_k)^\top A_{k'}(\alpha_{j_1} + \beta_0)\} + C_0, \tag{11}$$

We define $\phi$ as follows: for any Transformers

$$f_k = (\theta_k, \mathrm{pe}_k, (\mathbf{K}_{k;h}^{(l)}, \mathbf{Q}_{k;h}^{(l)}, \mathbf{V}_{k;h}^{(l)})_{h \in [H], l \in [L]}, \vartheta_k, \mathcal{V}_k),$$

over $\mathcal{V}_k$, $k \in [K]$, the Transformer $\widetilde{f} = \phi(f_1, \ldots, f_K)$ is given by

$$(\widetilde{\theta}, \widetilde{\mathrm{pe}}, (\widetilde{\mathbf{K}}_h^{(l)}, \widetilde{\mathbf{Q}}_h^{(l)}, \widetilde{\mathbf{V}}_h^{(l)})_{h \in [KH+1], l \in [L+1]}, \widetilde{\vartheta}, \mathcal{V}),$$

where the tokenizer is given by

$$\widetilde{\theta}(v) = \mathbb{1}(v \notin \mathcal{V}_0) \cdot \begin{pmatrix} \theta_1(v) \\ \vdots \\ \theta_K(v) \\ 0 \end{pmatrix} + \begin{pmatrix} 0 \\ \vdots \\ 0 \\ \beta_{\mathcal{E}(v)} \end{pmatrix}$$

where $\mathcal{E}(v) = k$ iff $v \in \mathcal{V}_k$. Let the positional encoder be given by

$$\widetilde{\mathrm{pe}}\left(\begin{pmatrix} x \\ y \end{pmatrix}; v_1, \ldots, v_i\right) = \begin{pmatrix} \mathrm{pe}_1(x; u) \\ \vdots \\ \mathrm{pe}_K(x; u) \\ \alpha_i + y \end{pmatrix},$$

where $x \in \mathbb{R}^d$ and $u$ is the sub-sequence of $v$ that omits $v_{i_0}$ (if any); for $l = 1, \ldots, L$ the key, query, value matrices are given by

$$\widetilde{\mathbf{K}}_{(k-1)H+h}^{(l)} = \begin{pmatrix} 0 & & & & \\ & \ddots & & & \\ & & \mathbf{K}_{k;h}^{(l)} & & \\ & & & \ddots & \\ & & & & A_0 \end{pmatrix}, \quad \widetilde{\mathbf{Q}}_{(k-1)H+h}^{(l)} = \begin{pmatrix} 0 & & & & \\ & \ddots & & & \\ & & \mathbf{Q}_{k;h}^{(l)} & & \\ & & & \ddots & \\ & & & & I \end{pmatrix},$$

$$\widetilde{\mathbf{V}}_{(k-1)H+h}^{(l)} = \begin{pmatrix} 0 & & & & \\ & \ddots & & & \\ & & \mathbf{V}_{k;h}^{(l)} & & \\ & & & \ddots & \\ & & & & 0 \end{pmatrix},$$

$$\widetilde{\mathbf{K}}_{KH+1}^{(l)} = \begin{pmatrix} 0 & & & \\ & \ddots & & \\ & & 0 & \\ & & & A \end{pmatrix}, \quad \widetilde{\mathbf{Q}}_{KH+1}^{(l)} = \begin{pmatrix} 0 & & & \\ & \ddots & & \\ & & 0 & \\ & & & I \end{pmatrix}, \quad \widetilde{\mathbf{V}}_{KH+1}^{(l)} = \begin{pmatrix} 0 & & & \\ & \ddots & & \\ & & 0 & \\ & & & I \end{pmatrix},$$

where the submatrices $\mathbf{K}_{k;h}^{(l)}, \mathbf{Q}_{k;h}^{(l)}, \mathbf{V}_{k;h}^{(l)}$ are located in the $k$-th diagonal block, and for the final layer

$$\widetilde{\mathbf{K}}_k^{(L+1)} = \begin{pmatrix} 0 \\ & \ddots \\ & & 0 \\ & & & A_k \end{pmatrix}, \widetilde{\mathbf{Q}}_k^{(L+1)} = \begin{pmatrix} 0 \\ & \ddots \\ & & 0 \\ & & & I \end{pmatrix}, \widetilde{\mathbf{V}}_k^{(L+1)} = \begin{pmatrix} 0 \\ & \ddots \\ & & I \\ & & & \ddots \\ & & & & 0 \end{pmatrix},$$

where the identity sub-matrix in $\widetilde{\mathbf{V}}_k^{(L+1)}$ is located in the $k$-th block. The output feature is given by

$\widetilde{\vartheta}(y) = \begin{pmatrix} \vartheta_1(y) \\ \vdots \\ \vartheta_K(y) \\ 0 \end{pmatrix}$. Since $u^{(k)}$'s only depend on set membership information of $v_i$'s, the general-

ized position encoding pe is well-defined. We can easily verify that $\phi$ is indeed a general-purpose Transformer of type $(O(K), O(\log N_{\max}))$.

We show that for any $l = 1, \dots, L$,

$$\widetilde{X}_i^{(l)} = \begin{pmatrix} X_{1;i}^{(l)} \\ \vdots \\ X_{K;i}^{(l)} \\ \widetilde{\alpha}_i \end{pmatrix}, \quad \forall i \neq i_0 \tag{12}$$

where $X_{k;i}^{(l)}$ is the $l$-th layer of Transformer $k$ at position $i$ (attending to all positions but $i_0$) such that

$$\|X_{k;i}^{(l)}\|_2 \leq B_\theta (KHB_v)^l. \tag{13}$$

and

$$\widetilde{X}_{i_0}^{(l)} = \begin{pmatrix} 0 \\ \vdots \\ 0 \\ \widetilde{\alpha}_{i_0} \end{pmatrix} \tag{14}$$

where $\widetilde{\alpha}_i = \alpha_i + \beta_{\mathcal{E}(v_i)}$.

We prove these results by induction. The case $l = 1$ folows directly from the definitions of the tokenizer.

**Prove Eq. (12).** Suppose Eq. (12) and Eq. (14) hold for $1, \dots, l-1$=th layer, and consider $l$-the layer. We have

$$\widetilde{X}_i^{(l+1)} = \underbrace{\sum_{k=1}^{K} \sum_{h=1}^{H} \sum_{j=1}^{i} \frac{\exp\left((\widetilde{\mathbf{Q}}_{(k-1)H+h}^{(l)} \widetilde{X}_i^{(l)})^\top (\widetilde{\mathbf{K}}_{(k-1)H+h}^{(l)} \widetilde{X}_j^{(l)})\right)}{\widetilde{Z}_{(k-1)H+h}^{(l)}} \cdot \widetilde{\mathbf{V}}_{(k-1)H+h}^{(l)} \widetilde{X}_j^{(l)}}_{\text{term 1}}$$

$$+ \underbrace{\sum_{j=1}^{i} \frac{\exp\left((\widetilde{\mathbf{Q}}_{KH+1}^{(l)} \widetilde{X}_i^{(l)})^\top (\widetilde{\mathbf{K}}_{KH+1}^{(l)} \widetilde{X}_j^{(l)})\right)}{\widetilde{Z}_{KH+1}^{(l)}} \cdot \widetilde{\mathbf{V}}_{KH+1}^{(l)} \widetilde{X}_j^{(l)}}_{\text{term 2}}.$$

Eq. (9) ensures that for any $j_1 < j_2 \leq i$ such that $i_0 \notin \{i, j_1, j_2\}$:

$$(\widetilde{\mathbf{Q}}_{(k-1)H+h}^{(l)} \widetilde{X}_i^{(l)})^\top (\widetilde{\mathbf{K}}_{(k-1)H+h}^{(l)} \widetilde{X}_{j_1}^{(l)}) = (\mathbf{Q}_{k;h}^{(l)} X_{k;i}^{(l)})^\top (\mathbf{K}_{k;h}^{(l)} X_{k;j_1}^{(l)}) + (\alpha_i + \beta_{\mathcal{E}(i)})^\top A_0 (\alpha_{j_1} + \beta_{\mathcal{E}(j_1)})$$

$$\geq (\mathbf{Q}_{k;h}^{(l)} X_{k;i}^{(l)})^\top (\mathbf{K}_{k;h}^{(l)} X_{k;j_1}^{(l)}) + (\alpha_i + \beta_{\mathcal{E}(i)})^\top A_0 (\alpha_{i_0} + \beta_{\mathcal{E}(i_0)}) + C$$

$$= (\widetilde{\mathbf{Q}}_{(k-1)H+h}^{(l)} \widetilde{X}_i^{(l)})^\top (\widetilde{\mathbf{K}}_{(k-1)H+h}^{(l)} \widetilde{X}_{i_0}^{(l)}) + C.$$

and

$$(\widetilde{\mathbf{Q}}^{(l)}_{(k-1)H+h}\widetilde{X}^{(l)}_i)^\top(\widetilde{\mathbf{K}}^{(l)}_{(k-1)H+h}\widetilde{X}^{(l)}_{j_1}) - (\widetilde{\mathbf{Q}}^{(l)}_{(k-1)H+h}\widetilde{X}^{(l)}_i)^\top(\widetilde{\mathbf{K}}^{(l)}_{(k-1)H+h}\widetilde{X}^{(l)}_{j_2})$$

$$= (\mathbf{Q}^{(l)}_{k;h}X^{(l)}_{k;i})^\top(\mathbf{K}^{(l)}_{k;h}X^{(l)}_{k;j_1}) + (\alpha_i + \beta_{\mathcal{E}(i)})^\top A_0(\alpha_{j_1} + \beta_{\mathcal{E}(j_1)})$$

$$- (\mathbf{Q}^{(l)}_{k;h}X^{(l)}_{k;i})^\top(\mathbf{K}^{(l)}_{k;h}X^{(l)}_{k;j_2}) - (\alpha_i + \beta_{\mathcal{E}(i)})^\top A_0(\alpha_{j_2} + \beta_{\mathcal{E}(j_2)})$$

$$= (\mathbf{Q}^{(l)}_{k;h}X^{(l)}_{k;i})^\top(\mathbf{K}^{(l)}_{k;h}X^{(l)}_{k;j_1}) - (\mathbf{Q}^{(l)}_{k;h}X^{(l)}_{k;i})^\top(\mathbf{K}^{(l)}_{k;h}X^{(l)}_{k;j_2}).$$

It follows from the precision $\epsilon$ of the transformers that the attention weights of head $(k-1)H + h$ is identical to the attention weights of expert $k$, i.e.

$$\frac{\exp\left((\widetilde{\mathbf{Q}}^{(l)}_{(k-1)H+h}\widetilde{X}^{(l)}_i)^\top(\widetilde{\mathbf{K}}^{(l)}_{(k-1)H+h}\widetilde{X}^{(l)}_j)\right)}{\widetilde{Z}^{(l)}_{(k-1)H+h}} = \frac{\exp\left((\mathbf{Q}^{(l)}_{k;h}X^{(l)}_{k;i})^\top(\mathbf{K}^{(l)}_{k;h}X^{(l)}_{k;j})\right)}{Z^{(l)}_{k;h}}.$$

Therefore

$$\text{term 1} = \sum_{k=1}^{K}\sum_{h=1}^{H}\sum_{j=1}^{i} \frac{\exp\left((\mathbf{Q}^{(l)}_{k;h}X^{(l)}_{k;i})^\top(\mathbf{K}^{(l)}_{k;h}X^{(l)}_{k;j})\right)}{Z^{(l)}_{k;h}} \cdot \begin{pmatrix} 0 \\ \vdots \\ \mathbf{V}^{(l)}_{k;h}X^{(l)}_{k;j} \\ \vdots \\ 0 \end{pmatrix} = \begin{pmatrix} X^{(l)}_{1;i} \\ \vdots \\ X^{(l)}_{K;i} \\ 0 \end{pmatrix}.$$

Furthermore, by Eq. (10) we have for any $j < i$

$$(\widetilde{\mathbf{Q}}^{(l)}_{KH+1}\widetilde{X}^{(l)}_i)^\top(\widetilde{\mathbf{K}}^{(l)}_{KH+1}\widetilde{X}^{(l)}_i) = \widetilde{\alpha}_i^\top A \widetilde{\alpha}_i$$

$$\geq \widetilde{\alpha}_i^\top A \widetilde{\alpha}_j + C$$

$$= (\widetilde{\mathbf{Q}}^{(l)}_{KH+1}\widetilde{X}^{(l)}_i)^\top(\widetilde{\mathbf{K}}^{(l)}_{KH+1}\widetilde{X}^{(l)}_j) + C$$

and hence the attention weighs concentrates on $i$ itself. Thus

$$\text{term 2} = \begin{pmatrix} 0 & & & \\ & \ddots & & \\ & & 0 & \\ & & & I \end{pmatrix} \cdot \begin{pmatrix} X^{(l)}_{1;i} \\ \vdots \\ X^{(l)}_{K;i} \\ \widetilde{\alpha}_i \end{pmatrix} = \begin{pmatrix} 0 \\ \vdots \\ 0 \\ \widetilde{\alpha}_i \end{pmatrix}.$$

Combining these two terms, we confirm that Eq.(12) holds for $(l+1)$-th layer.

**Prove Eq. (13).** From above,

$$\|X^{(l+1)}_{k;i}\|_2 = \left\|\sum_{k=1}^{K}\sum_{h=1}^{H}\sum_{j=1}^{i} \frac{\exp\left((\widetilde{\mathbf{Q}}^{(l)}_{(k-1)H+h}\widetilde{X}^{(l)}_i)^\top(\widetilde{\mathbf{K}}^{(l)}_{(k-1)H+h}\widetilde{X}^{(l)}_j)\right)}{\widetilde{Z}^{(l)}_{(k-1)H+h}} \cdot \mathbf{V}^{(l)}_{k;h}X^{(l)}_{k;j}\right\|_2$$

$$\leq KHB_v \cdot \max_{j \leq i}\|X^{(l)}_{k;j}\|_2$$

$$\leq B_\theta(KHB_v)^{l+1}.$$

This confirms Eq. (13) for $l+1$.

**Prove Eq. (14).** Notice that Eq. (9) ensures that for any $j \leq i_0$:

$$(\widetilde{\mathbf{Q}}^{(l)}_{(k-1)H+h}\widetilde{X}^{(l)}_{i_0})^\top(\widetilde{\mathbf{K}}^{(l)}_{(k-1)H+h}\widetilde{X}^{(l)}_{i_0}) = (\mathbf{Q}^{(l)}_{k;h}X^{(l)}_{k;i_0})^\top(\mathbf{K}^{(l)}_{k;h}X^{(l)}_{k;i_0}) + (\alpha_{i_0} + \beta_{\mathcal{E}(i_0)})^\top A_0(\alpha_{i_0} + \beta_{\mathcal{E}(i_0)})$$

$$\geq (\mathbf{Q}^{(l)}_{k;h}X^{(l)}_{k;i_0})^\top(\mathbf{K}^{(l)}_{k;h}X^{(l)}_{k;j}) + (\alpha_{i_0} + \beta_{\mathcal{E}(i_0)})^\top A_0(\alpha_j + \beta_{\mathcal{E}(j)}) + C$$

$$= (\widetilde{\mathbf{Q}}^{(l)}_{(k-1)H+h}\widetilde{X}^{(l)}_{i_0})^\top(\widetilde{\mathbf{K}}^{(l)}_{(k-1)H+h}\widetilde{X}^{(l)}_j) + C.$$

It follows that the attention weights of head $(k-1)H + h$ is concentrated on $i_0$ itself, therefore

$$\text{term 1} = \sum_{k=1}^{K} \sum_{h=1}^{H} \begin{pmatrix} 0 \\ \vdots \\ \mathbf{V}_{k;h}^{(l)} \cdot 0 \\ \vdots \\ 0 \end{pmatrix} = 0.$$

By the same argument, for $i = i_0$ we have

$$\text{term 2} = \begin{pmatrix} 0 & & \\ & \ddots & \\ & & 0 \\ & & I \end{pmatrix} \cdot \begin{pmatrix} 0 \\ \vdots \\ 0 \\ \widetilde{\alpha}_{i_0} \end{pmatrix} = \begin{pmatrix} 0 \\ \vdots \\ 0 \\ \widetilde{\alpha}_{i_0} \end{pmatrix}.$$

Combining these confirms Eq. (14).

Next, we show that the last layer satisfies

$$\widetilde{X}_n^{(L+1)} = \begin{pmatrix} 0 \\ \vdots \\ X_{\kappa;n}^{(L+1)} \\ \vdots \\ 0 \end{pmatrix} \tag{15}$$

where $X_{\kappa;n}^{(L+1)}$ is the $\kappa$-th block. To see this, we notice that Eq. (11) implies the followings (the proofs are identical to the above):

1. Attention sink to dummy token $v_{i_0}$ for mismatch expert: for any $k' \neq \kappa$ and $j \leq n$ we have
$$\begin{aligned}
(\widetilde{\mathbf{Q}}_{(k'-1)H+h}^{(L)} \widetilde{X}_n^{(L)})^\top (\widetilde{\mathbf{K}}_{(k'-1)H+h}^{(L)} \widetilde{X}_j^{(L)}) &= (\alpha_n + \beta_{\mathcal{E}(n)})^\top A_{k'} (\alpha_j + \beta_{\mathcal{E}(j)}) \\
&\leq (\alpha_n + \beta_{\mathcal{E}(n)})^\top A_{k'} (\alpha_{i_0} + \beta_{\mathcal{E}(i_0)}) - C \\
&= (\widetilde{\mathbf{Q}}_{(k'-1)H+h}^{(L)} \widetilde{X}_n^{(L)})^\top (\widetilde{\mathbf{K}}_{(k'-1)H+h}^{(L)} \widetilde{X}_{i_0}^{(L)}) - C.
\end{aligned} \tag{16}$$

2. Attention to oneself for matching expert: for any $j \neq i_0$ we have
$$\begin{aligned}
(\widetilde{\mathbf{Q}}_{(\kappa-1)H+h}^{(L)} \widetilde{X}_n^{(L)})^\top (\widetilde{\mathbf{K}}_{(\kappa-1)H+h}^{(L)} \widetilde{X}_j^{(L)}) &= (\alpha_n + \beta_{\mathcal{E}(n)})^\top A_\kappa (\alpha_j + \beta_{\mathcal{E}(j)}) \\
&\geq (\alpha_n + \beta_{\mathcal{E}(n)})^\top A_\kappa (\alpha_{i_0} + \beta_{\mathcal{E}(i_0)}) + C \\
&= (\widetilde{\mathbf{Q}}_{(\kappa-1)H+h}^{(L)} \widetilde{X}_n^{(L)})^\top (\widetilde{\mathbf{K}}_{(\kappa-1)H+h}^{(L)} \widetilde{X}_{i_0}^{(L)}) + C,
\end{aligned} \tag{17}$$

and
$$\begin{aligned}
(\widetilde{\mathbf{Q}}_{(\kappa-1)H+h}^{(L)} \widetilde{X}_n^{(L)})^\top (\widetilde{\mathbf{K}}_{(\kappa-1)H+h}^{(L)} \widetilde{X}_n^{(L)}) &= (\alpha_n + \beta_{\mathcal{E}(n)})^\top A_\kappa (\alpha_n + \beta_{\mathcal{E}(n)}) \\
&\geq (\alpha_n + \beta_{\mathcal{E}(n)})^\top A_\kappa (\alpha_j + \beta_{\mathcal{E}(j)}) + C \\
&= (\widetilde{\mathbf{Q}}_{(\kappa-1)H+h}^{(L)} \widetilde{X}_n^{(L)})^\top (\widetilde{\mathbf{K}}_{(\kappa-1)H+h}^{(L)} \widetilde{X}_j^{(L)}) + C.
\end{aligned} \tag{18}$$

Combining Eq. (16), Eq. (17), and Eq. (18), we have

$$\frac{\exp\left((\widetilde{\mathbf{Q}}_{(k-1)H+h}^{(L)} \widetilde{X}_n^{(L)})^\top (\widetilde{\mathbf{K}}_{(k-1)H+h}^{(L)} \widetilde{X}_j^{(L)})\right)}{Z_k^{(l)}} = \begin{cases} \delta_j^{i_0}, & k \neq \kappa \\ \delta_j^n, & k = \kappa \end{cases}$$

It follows that

$$\widetilde{X}_n^{(L+1)} = \widetilde{\mathbf{V}}_{(\kappa-1)H+h}^{(L)} \cdot \widetilde{X}_n^{(L)} + \sum_{k \neq \kappa} \mathbf{V}_{(\kappa-1)H+h}^{(L)} \cdot \widetilde{X}_{i_0}^{(L)}$$

$$= \begin{pmatrix} 0 & & & & \\ & \ddots & & & \\ & & I & & \\ & & & \ddots & \\ & & & & 0 \end{pmatrix} \cdot \begin{pmatrix} X_{1;i}^{(L)} \\ \vdots \\ X_{K;i}^{(L)} \\ \widetilde{\alpha}_i \end{pmatrix} = \begin{pmatrix} 0 \\ \vdots \\ X_{\kappa;n}^{(L)} \\ \vdots \\ 0 \end{pmatrix}.$$

Therefore we establish Eq. (15).

Finally, at the output layer

$$p_{\widetilde{f}}(y|v_1, \ldots, v_n) = \mathrm{Softmax}(\widetilde{\vartheta}(y)^\top \widetilde{X}_n^{(L+1)})$$

$$= \mathrm{Softmax}(\vartheta(y)^\top Y_{n-1}^{(L)})$$

$$= p_{f_\kappa}(y|u).$$

This establishes the desired statement. $\qquad\square$

### A.4 PROOF OF PROPOSITION 4.4

*Proof.* Set constants $B_v, B_{qk}, B_\theta$ such that for any layer $l$ and head $h$, it holds that $\left\|(\mathbf{Q}_h^{(l)})^\top \mathbf{K}_h^{(l)}\right\|_2 \leq B_{qk}$, $\left\|\mathbf{V}_h^{(l)}\right\|_2 \leq B_v$, and $\|\theta(v)\|_2 \leq B_\theta$ holds for all $v \in \mathcal{V}$. Let $B = (KHB_v)^L B_{qk} B_\theta, C = 2B^2 + \log(1/\epsilon), C_0 = 4C$. Define $\iota(i) = u$ iff $\xi_u \leq i < \xi_{u+1}$ ($\xi_0 = -1, \xi_{m+1} = \infty$ by default). Let $\mathcal{E}(\cdot)$ denote the task id indicated by the special token. By Lemma A.2, there exists $\alpha_1, \ldots, \alpha_N, \beta_1, \ldots, \beta_K \in \mathbb{R}^{d_0}$ and $A, A_1, \ldots, A_K \in \mathbb{R}^{d_0 \times d_0}$ for $d_0 \leq O(K + \log N_{\max})$ such that for any $n \leq N$ we have

1. For any $k \neq k'$:
$$\alpha_n^\top A_k(\alpha_n + \beta_{k'}) \geq C_0 + \begin{cases} \alpha_n^\top A_k \alpha_n \\ \alpha_n^\top A_k \alpha_j \\ \alpha_n^\top A_k(\alpha_j + \beta_{k''}) \end{cases} , \forall 0 \leq j < n, 1 \leq k'' \leq K. \quad (19)$$

2. For any $k \in [K]$:
$$\alpha_n^\top A_k \alpha_n = \alpha_n^\top A_k \alpha_0 \geq C_0 + \begin{cases} \alpha_n^\top A_k(\alpha_n + \beta_k) \\ \alpha_n^\top A_k \alpha_j \\ \alpha_n^\top A_k(\alpha_j + \beta_{k'}) \end{cases} , \forall 0 < j < n, k' \neq k. \quad (20)$$

3. For any $k, k', k'' \in [K]$:
$$(\alpha_n + \beta_{k'})^\top A_k(\alpha_n + \beta_{k'}) \geq C_0 + (\alpha_n + \beta_{k'})^\top A_k \alpha_j, \forall 0 \leq j \leq n. \quad (21)$$

4. For any $0 < j < n$:
$$\alpha_n^\top A \alpha_n \geq \alpha_n^\top A(\alpha_n + \beta_k) + C_0$$
$$\geq C_0 + \max\{\alpha_n^\top A \alpha_j, \alpha_n^\top A(\alpha_j + \beta_{k'})\}, \forall k, k'' \in [K]. \quad (22)$$

We define $\phi$ as follows: for any Transformers
$$f_k = (\theta_k, \mathrm{pe}_k, (\mathbf{K}_{k;h}^{(l)}, \mathbf{Q}_{k;h}^{(l)}, \mathbf{V}_{k;h}^{(l)})_{h \in [H], l \in [L]}, \vartheta_k, \mathcal{V}), k \in [K]$$
over $\mathcal{V}$, the Transformer $\widetilde{f} = \phi(f_1, \ldots, f_K)$ is given by
$$(\widetilde{\theta}, \widetilde{\mathrm{pe}}, (\widetilde{\mathbf{K}}_h^{(l)}, \widetilde{\mathbf{Q}}_h^{(l)}, \widetilde{\mathbf{V}}_h^{(l)})_{h \in [KH+1], l \in [L]}, \widetilde{\vartheta}, \mathcal{V} \cup \Omega),$$

where the tokenizer is given by

$$\widetilde{\theta}(v) = \begin{pmatrix} \theta_1(v) \\ \vdots \\ \theta_K(v) \\ 0 \end{pmatrix}, \ v \in \mathcal{V}, \ \widetilde{\theta}(\omega) = \begin{pmatrix} 0 \\ \vdots \\ 0 \\ \beta_{\mathcal{E}(\omega)} \end{pmatrix}, \ \omega \in \Omega,$$

the positional encoder is given by

$$\widetilde{\text{pe}}\left(\begin{pmatrix} x \\ y \end{pmatrix}; v_1, \ldots, v_i\right) = \begin{pmatrix} \text{pe}_1\left(x; v_1, \cdots, v_{\xi_1 - 1}, v_{\xi_{\iota(i)} + 1}, \cdots, v_i\right) \\ \vdots \\ \text{pe}_K\left(x; v_1, \cdots, v_{\xi_1 - 1}, v_{\xi_{\iota(i)} + 1}, \cdots, v_i\right) \\ \alpha_{\iota(i)} + y \end{pmatrix},$$

where $x \in \mathbb{R}^d$; for $l = 1, \ldots, L$ the key, query, value matrices are given by

$$\widetilde{\mathbf{K}}^{(l)}_{(k-1)H+h} = \begin{pmatrix} 0 & & & & \\ & \ddots & & & \\ & & \mathbf{K}^{(l)}_{k;h} & & \\ & & & \ddots & \\ & & & & A_k \end{pmatrix}, \ \widetilde{\mathbf{Q}}^{(l)}_{(k-1)H+h} = \begin{pmatrix} 0 & & & & \\ & \ddots & & & \\ & & \mathbf{Q}^{(l)}_{k;h} & & \\ & & & \ddots & \\ & & & & I \end{pmatrix},$$

$$\widetilde{\mathbf{V}}^{(l)}_{(k-1)H+h} = \begin{pmatrix} 0 & & & & \\ & \ddots & & & \\ & & \mathbf{V}^{(l)}_{k;h} & & \\ & & & \ddots & \\ & & & & 0 \end{pmatrix},$$

$$\widetilde{\mathbf{K}}^{(l)}_{KH+1} = \begin{pmatrix} 0 & & & \\ & \ddots & & \\ & & 0 & \\ & & & A \end{pmatrix}, \ \widetilde{\mathbf{Q}}^{(l)}_{KH+1} = \begin{pmatrix} 0 & & & \\ & \ddots & & \\ & & 0 & \\ & & & I \end{pmatrix}, \ \widetilde{\mathbf{V}}^{(l)}_{KH+1} = \begin{pmatrix} 0 & & & \\ & \ddots & & \\ & & 0 & \\ & & & I \end{pmatrix},$$

where the submatrices $\mathbf{K}^{(l)}_{k;h}, \mathbf{Q}^{(l)}_{k;h}, \mathbf{V}^{(l)}_{k;h}$ are located in the $k$-th diagonal block. The output feature is given by $\widetilde{\vartheta}(y) = \begin{pmatrix} \vartheta_1(y) \\ \vdots \\ \vartheta_K(y) \\ 0 \end{pmatrix}$. Since $\xi_1, \xi_m$ only depends on whether $v_i$'s belong to the set $\Omega$, the generalized position encoding pe is well-defined. We can easily verify that $\phi$ is indeed a general-purpose Transformer of type $(O(K), O(\log N_{\max}))$.

Let $\widetilde{X}^{(l)}_1, \ldots, \widetilde{X}^{(l)}_n$ represent the $l$-th hidden layer. Our goal is to show that for any $l = 1, \ldots, L$, $\widetilde{X}^{(l)}_i$ can be written as:

$$\widetilde{X}^{(l)}_i = \begin{pmatrix} X^{(l)}_{1;i} \\ \vdots \\ X^{(l)}_{K;i} \\ \widetilde{\alpha}_i \end{pmatrix}, \ i = 1, \ldots, n, \tag{23}$$

where $\widetilde{\alpha}_i = \alpha_{\iota(i)} + \mathbb{1}(v_i \in \Omega) \cdot \beta_{\mathcal{E}(v_i)}$ and $X^{(l)}_{k;i} \in \mathbb{R}^d$ such that

$$\|X^{(l)}_{k;i}\|_2 \leq B_\theta (KHB_v)^l. \tag{24}$$

In particular, for $i = 1, \ldots, m$ we have

$$X^{(l)}_{k;\xi_i} = 0, \ \forall k = 1, \ldots, K, \tag{25}$$

and for $j = 1, \dots, \xi_1 - 1$ we have

$$X_{k;j}^{(l)} = Y_{k;j}^{(l)}, \ \forall k = 1, \dots, K, \tag{26}$$

and for $j = 1, \dots, \xi_1 - 1, \xi_m + 1, \dots, n$ we have

$$X_{\kappa;j}^{(l)} = Y_{\kappa, j - \xi_m - 1 + \xi_1}^{(l)}, \ X_{k';j}^{(l)} = 0, \ \forall k' \neq \kappa, \tag{27}$$

where $Y_{k;j}^{(l)}$ is the $l$-th hidden layer of $f_k$ (attending only to positions $1, \dots, \xi_1 - 1, \xi_m + 1, \dots, n$).

Thus we apply induction on $l$. The case $l = 1$ holds trivially from the definition of $\widetilde{\theta}$ and $\widetilde{\text{pe}}$. Suppose the above relationship holds for all layers $1, \dots, l$, consider layer $l + 1$. We have

$$
\widetilde{X}_i^{(l+1)} = \underbrace{\sum_{k=1}^{K} \sum_{h=1}^{H} \sum_{j=1}^{i} \frac{\exp\left( (\widetilde{\mathbf{Q}}_{(k-1)H+h}^{(l)} \widetilde{X}_i^{(l)})^\top (\widetilde{\mathbf{K}}_{(k-1)H+h}^{(l)} \widetilde{X}_j^{(l)}) \right)}{\widetilde{Z}_{(k-1)H+h}^{(l)}} \cdot \widetilde{\mathbf{V}}_{(k-1)H+h}^{(l)} \widetilde{X}_j^{(l)}}_{\text{term 1}}
$$

$$
+ \underbrace{\sum_{j=1}^{i} \frac{\exp\left( (\widetilde{\mathbf{Q}}_{KH+1}^{(l)} \widetilde{X}_i^{(l)})^\top (\widetilde{\mathbf{K}}_{KH+1}^{(l)} \widetilde{X}_j^{(l)}) \right)}{\widetilde{Z}_{KH+1}^{(l)}} \cdot \widetilde{\mathbf{V}}_{KH+1}^{(l)} \widetilde{X}_j^{(l)}}_{\text{term 2}},
$$

where

$$
\widetilde{Z}_{(k-1)H+h}^{(l)} = \sum_{j=1}^{i} \exp\left( (\widetilde{\mathbf{Q}}_{(k-1)H+h}^{(l)} \widetilde{X}_i^{(l)})^\top (\widetilde{\mathbf{K}}_{(k-1)H+h}^{(l)} \widetilde{X}_j^{(l)}) \right).
$$

By induction hypothesis,

$$
\widetilde{X}_i^{(l)} = \begin{pmatrix} X_{1;i}^{(l)} \\ \vdots \\ X_{K;i}^{(l)} \\ \widetilde{\alpha}_i \end{pmatrix},
$$

and $X_{k;i}^{(l)} = Y_{\zeta(i)}^{(l)}$ for $i = 1, \dots, \xi_1 - 1, \xi_m + 1, \dots, n$, where $\zeta(i) := \begin{cases} i, & i < \xi_1 \\ i - \xi_m - 1 + \xi_1, & i > \xi_m \end{cases}$.

Notice that for $j \leq i$:

$$(\widetilde{\mathbf{Q}}_{(k-1)H+h}^{(l)} \widetilde{X}_i^{(l)})^\top (\widetilde{\mathbf{K}}_{(k-1)H+h}^{(l)} \widetilde{X}_j^{(l)}) = (X_{k;i}^{(l)})^\top (\mathbf{Q}_{k;h}^{(l)})^\top \mathbf{K}_{k;h}^{(l)} X_{k;j}^{(l)} + \widetilde{\alpha}_i^\top A_k \widetilde{\alpha}_j,$$

$$(\widetilde{\mathbf{Q}}_{KH+1}^{(l)} \widetilde{X}_i^{(l)})^\top (\widetilde{\mathbf{K}}_{KH+1}^{(l)} \widetilde{X}_j^{(l)}) = \widetilde{\alpha}_i^\top A \widetilde{\alpha}_j.$$

**Prove Eq (23).** By properties of $\alpha, \beta, A$, for any $j_2 < \xi_u < j_1 < i < \xi_{u+1}$ notice that:

$$(\widetilde{\mathbf{Q}}_{KH+1}^{(l)} \widetilde{X}_i^{(l)})^\top (\widetilde{\mathbf{K}}_{KH+1}^{(l)} \widetilde{X}_{j_1}^{(l)}) \geq (\widetilde{\mathbf{Q}}_{KH+1}^{(l)} \widetilde{X}_i^{(l)})^\top (\widetilde{\mathbf{K}}_{KH+1}^{(l)} \widetilde{X}_{\xi_u}^{(l)}) + C$$

$$\geq (\widetilde{\mathbf{Q}}_{KH+1}^{(l)} \widetilde{X}_i^{(l)})^\top (\widetilde{\mathbf{K}}_{KH+1}^{(l)} \widetilde{X}_{j_2}^{(l)}) + 2C.$$

Due to $\epsilon$-precision of transformers, this implies that

$$
\frac{\exp\left( (\widetilde{\mathbf{Q}}_{KH+1}^{(l)} \widetilde{X}_i^{(l)})^\top (\widetilde{\mathbf{K}}_{KH+1}^{(l)} \widetilde{X}_j^{(l)}) \right)}{Z_{KH+1}^{(l)}} = \begin{cases} \frac{\mathbb{1}(j > \xi_u)}{i - \xi_u}, & \xi_u < i < \xi_{u+1} \\ \delta_{\xi_l}^j, & i = \xi_u \end{cases},
$$

and hence for $\xi_u < i < \xi_{u+1}$

$$\widetilde{X}_i^{(l+1)} = \sum_{k=1}^K \sum_{h=1}^H \sum_{j=1}^i \frac{\exp\left((\widetilde{\mathbf{Q}}_{(k-1)H+h}^{(l)} \widetilde{X}_i^{(l)})^\top (\widetilde{\mathbf{K}}_{(k-1)H+h}^{(l)} \widetilde{X}_j^{(l)})\right)}{\widetilde{Z}_{(k-1)H+h}^{(l)}} \cdot \widetilde{\mathbf{V}}_{(k-1)H+h}^{(l)} \begin{pmatrix} \vdots \\ X_{k;j}^{(l)} \\ \vdots \\ 0 \end{pmatrix}$$

$$+ \sum_{j=\xi_u+1}^i \cdot \frac{1}{i-\xi_u} \cdot \begin{pmatrix} 0 \\ \vdots \\ 0 \\ \alpha_{\iota(i)} \end{pmatrix}$$

$$= \begin{pmatrix} X_{1;i}^{(l+1)} \\ \vdots \\ X_{K;i}^{(l+1)} \\ \widetilde{\alpha}_i \end{pmatrix},$$

and for $i = \xi_u$

$$\widetilde{X}_i^{(l+1)} = \sum_{k=1}^K \sum_{h=1}^H \sum_{j=1}^i \frac{\exp\left((\widetilde{\mathbf{Q}}_{(k-1)H+h}^{(l)} \widetilde{X}_i^{(l)})^\top (\widetilde{\mathbf{K}}_{(k-1)H+h}^{(l)} \widetilde{X}_j^{(l)})\right)}{\widetilde{Z}_{(k-1)H+h}^{(l)}} \cdot \widetilde{\mathbf{V}}_{(k-1)H+h}^{(l)} \begin{pmatrix} \vdots \\ X_{k;j}^{(l)} \\ \vdots \\ 0 \end{pmatrix} + \begin{pmatrix} 0 \\ \vdots \\ 0 \\ \alpha_{\iota(i)} + \beta_{\mathcal{E}(v_i)} \end{pmatrix}$$

$$= \begin{pmatrix} X_{1;i}^{(l+1)} \\ \vdots \\ X_{K;i}^{(l+1)} \\ \widetilde{\alpha}_i \end{pmatrix},$$

where

$$X_{k;i}^{(l+1)} = \sum_{k=1}^K \sum_{h=1}^H \sum_{j=1}^i \frac{\exp\left((\widetilde{\mathbf{Q}}_{(k-1)H+h}^{(l)} \widetilde{X}_i^{(l)})^\top (\widetilde{\mathbf{K}}_{(k-1)H+h}^{(l)} \widetilde{X}_j^{(l)})\right)}{\widetilde{Z}_{(k-1)H+h}^{(l)}} \cdot \mathbf{V}_{k;h}^{(l)} X_{k;j}^{(l)}. \tag{28}$$

This confirms Eq. (23) for $l+1$.

**Prove Eq. (24).** From above,

$$\|X_{k;i}^{(l+1)}\|_2 = \left\|\sum_{k=1}^K \sum_{h=1}^H \sum_{j=1}^i \frac{\exp\left((\widetilde{\mathbf{Q}}_{(k-1)H+h}^{(l)} \widetilde{X}_i^{(l)})^\top (\widetilde{\mathbf{K}}_{(k-1)H+h}^{(l)} \widetilde{X}_j^{(l)})\right)}{\widetilde{Z}_{(k-1)H+h}^{(l)}} \cdot \mathbf{V}_{k;h}^{(l)} X_{k;j}^{(l)}\right\|_2$$

$$\leq KHB_v \cdot \max_{j \leq i} \|X_{k;j}^{(l)}\|_2$$

$$\leq B_\theta (KHB_v)^{l+1}.$$

This confirms Eq. (24) for $l+1$.

**Prove Eq. (25).** We first show $X_{k;\xi_1}^{(l)} = 0$. Indeed, by the properties of $\alpha_t, \beta_k$, for any $j \leq \xi_1$

$$(\widetilde{\mathbf{Q}}_{(k-1)H+h}^{(l)} \widetilde{X}_{\xi_1}^{(l)})^\top (\widetilde{\mathbf{K}}_{(k-1)H+h}^{(l)} \widetilde{X}_{\xi_1}^{(l)})$$

$$= (X_{k;\xi_1}^{(l)})^\top (\mathbf{Q}_{k;h}^{(l)})^\top \mathbf{K}_{k;h}^{(l)} X_{k;\xi_1}^{(l)} + (\alpha_0 + \beta_{\mathcal{E}(v_{\xi_1})})^\top A_k (\alpha_0 + \beta_{\mathcal{E}(v_{\xi_1})})$$

$$\geq (X_{k;\xi_1}^{(l)})^\top (\mathbf{Q}_{k;h}^{(l)})^\top \mathbf{K}_{k;h}^{(l)} X_{k;\xi_1}^{(l)} + (\alpha_0 + \beta_{\mathcal{E}(v_{\xi_1})})^\top A_k \alpha_0 + C$$

$$= (\widetilde{\mathbf{Q}}_{(k-1)H+h}^{(l)} \widetilde{X}_{\xi_1}^{(l)})^\top (\widetilde{\mathbf{K}}_{(k-1)H+h}^{(l)} \widetilde{X}_j^{(l)}) + C$$

It follows from Eq. (28) that

$$X_{k;\xi_1}^{(l+1)} = \sum_{k=1}^{K} \sum_{h=1}^{H} \mathbf{V}_{k;h}^{(l)} X_{k;\xi_1}^{(l)} = 0.$$

For $\xi_i$ $(i > 1)$, we apply the same argument again to obtain that for any $j \le \xi_i$ such that $j \notin \{\xi_1 < \cdots < \xi_{\iota(n)}\}$ and any $i' < i$,

$$(\widetilde{\mathbf{Q}}_{(k-1)H+h}^{(l)} \widetilde{X}_{\xi_i}^{(l)})^\top (\widetilde{\mathbf{K}}_{(k-1)H+h}^{(l)} \widetilde{X}_{\xi_{k'}}^{(l)})$$
$$\ge (\widetilde{\mathbf{Q}}_{(k-1)H+h}^{(l)} \widetilde{X}_{\xi_1}^{(l)})^\top (\widetilde{\mathbf{K}}_{(k-1)H+h}^{(l)} \widetilde{X}_{j}^{(l)}) + C$$

This implies that the attention weights are supported on $\{\xi_1 < \cdots < \xi_i\}$, and therefore

$$X_{k;\xi_i}^{(l+1)} = \sum_{k=1}^{K} \sum_{h=1}^{H} \sum_{j=1}^{i} \frac{\exp\left((\widetilde{\mathbf{Q}}_{(k-1)H+h}^{(l)} \widetilde{X}_{\xi_i}^{(l)})^\top (\widetilde{\mathbf{K}}_{(k-1)H+h}^{(l)} \widetilde{X}_{\xi_j}^{(l)})\right)}{\widetilde{Z}_{(k-1)H+h}^{(l)}} \cdot \mathbf{V}_{k;h}^{(l)} X_{k;\xi_j}^{(l)} = 0$$

where we apply the induction hypothesis $k; X_{\xi_j}^{(l)} = 0$ for all $j = 1, \ldots, i-1$. This thus completes the proof of Eq. (25).

**Prove Eq. (26).** When $j_1 < j_2 \le i < \xi_1$, we have

$$(\widetilde{\mathbf{Q}}_{(k-1)H+h}^{(l)} \widetilde{X}_{i}^{(l)})^\top (\widetilde{\mathbf{K}}_{(k-1)H+h}^{(l)} \widetilde{X}_{j_1}^{(l)}) - (\widetilde{\mathbf{Q}}_{(k-1)H+h}^{(l)} \widetilde{X}_{i}^{(l)})^\top (\widetilde{\mathbf{K}}_{(k-1)H+h}^{(l)} X_{j_2}^{(l)})$$
$$= (X_{k;i}^{(l)})^\top (\mathbf{Q}_{k;h}^{(l)})^\top \mathbf{K}_{k;h}^{(l)} X_{k;j_1}^{(l)} + \alpha_0^\top A_k \alpha_0^\top$$
$$- (X_{k;i}^{(l)})^\top (\mathbf{Q}_{k;h}^{(l)})^\top \mathbf{K}_{k;h}^{(l)} X_{k;j_2}^{(l)} - \alpha_0^\top A_k \alpha_0^\top$$
$$= (\mathbf{Q}_{k;h}^{(l)} Y_{k;i}^{(l)})^\top (\mathbf{K}_{k;h}^{(l)} Y_{k;j_i}^{(l)}) - (\mathbf{Q}_{k;h}^{(l)} Y_{k;i}^{(l)})^\top (\mathbf{K}_{k;h}^{(l)} Y_{k;j_2}^{(l)}).$$

It follows that

$$\widetilde{Z}_{(k-1)H+h}^{(l)} = \sum_{j=1}^{i} \exp\left((\mathbf{Q}_{k;h}^{(l)} Y_{k;i}^{(l)})^\top (\mathbf{K}_{k;h}^{(l)} Y_{k;j}^{(l)})\right),$$

and

$$X_{k;i}^{(l+1)} = \sum_{k=1}^{K} \sum_{h=1}^{H} \sum_{j=1}^{i} \frac{\exp\left((\mathbf{Q}_{k;h}^{(l)} Y_{k;i}^{(l)})^\top (\mathbf{K}_{k;h}^{(l)} Y_{k;j}^{(l)})\right)}{\widetilde{Z}_{(k-1)H+h}^{(l)}} \cdot \mathbf{V}_{k;h}^{(l)} Y_{k;j}^{(l)}$$
$$= Y_{k;i}^{(l+1)}.$$

This confirms Eq. (26).

**Prove Eq. (27).** When $i > \xi_m$, we rely on the following properties:

1. Attention sink to $v_{\xi_m}$ for mismatch expert: for any $k' \ne \kappa$ and $j \le i$ we have
$$(\widetilde{\mathbf{Q}}_{(k'-1)H+h}^{(l)} \widetilde{X}_{i}^{(l)})^\top (\widetilde{\mathbf{K}}_{(k'-1)H+h}^{(l)} \widetilde{X}_{j}^{(l)}) \le (\widetilde{\mathbf{Q}}_{(k'-1)H+h}^{(l)} \widetilde{X}_{i}^{(l)})^\top (\widetilde{\mathbf{K}}_{(k'-1)H+h}^{(l)} \widetilde{X}_{\xi_m}^{(l)}) - C.$$
(29)

2. Attention to task-relevant tokens for matching expert: for $j \in \{1, \ldots, \xi_1 - 1, \xi_m + 1, \ldots, n\}$, and $\xi_1 \le j' \le \xi_m$ we have
$$(\widetilde{\mathbf{Q}}_{(\kappa-1)H+h}^{(l)} \widetilde{X}_{i}^{(l)})^\top (\widetilde{\mathbf{K}}_{(\kappa-1)H+h}^{(l)} \widetilde{X}_{j}^{(l)}) \ge (\widetilde{\mathbf{Q}}_{(\kappa-1)H+h}^{(l)} \widetilde{X}_{i}^{(l)})^\top (\widetilde{\mathbf{K}}_{(\kappa-1)H+h}^{(l)} \widetilde{X}_{j'}^{(l)}) + C.$$
(30)

and for $j_1 < j_2 \in \{1, \ldots, \xi - 1 - 1, \xi_m + 1, \ldots, n\}$
$$(\widetilde{\mathbf{Q}}_{(\kappa-1)H+h}^{(l)} \widetilde{X}_{i}^{(l)})^\top (\widetilde{\mathbf{K}}_{(\kappa-1)H+h}^{(l)} \widetilde{X}_{j_1}^{(l)}) - (\widetilde{\mathbf{Q}}_{(\kappa-1)H+h}^{(l)} \widetilde{X}_{i}^{(l)})^\top (\widetilde{\mathbf{K}}_{(\kappa-1)H+h}^{(l)} \widetilde{X}_{j_2}^{(l)})$$
$$= (\mathbf{Q}_{\kappa;h}^{(l)} Y_{\kappa;i-\xi_m-1+\xi_1}^{(l)})^\top (\mathbf{K}_{\kappa;h}^{(l)} Y_{\zeta(j_1)}^{(l)}) - (\mathbf{Q}_{\kappa;h}^{(l)} Y_{i-\xi_m-1+\xi_1}^{(l)})^\top \mathbf{K}_{\kappa;h}^{(l)} Y_{\kappa;\zeta(j_2)}^{(l)}),$$
(31)

To see Eq. (29), we notice that

$$(\widetilde{\mathbf{Q}}_{(k'-1)H+h}^{(l)} \widetilde{X}_i^{(l)})^\top (\widetilde{\mathbf{K}}_{(k'-1)H+h}^{(l)} \widetilde{X}_j^{(l)})$$

$$= (X_{k';i}^{(l)})^\top (\mathbf{Q}_{k';h}^{(l)})^\top \mathbf{K}_{k';h}^{(l)} X_{k';j}^{(l)} + \alpha_m^\top A_{k'}(\alpha_{\iota(j)} + \beta_{\mathcal{E}(v_j)} \cdot \mathbb{1}(v_j \in \Omega))$$

$$\leq (X_{k';i}^{(l)})^\top (\mathbf{Q}_{k';h}^{(l)})^\top \mathbf{K}_{k';h}^{(l)} X_{k';\xi_m}^{(l)} + \alpha_m^\top A_{k'}(\alpha_m + \beta_{\mathcal{E}(v_{\xi_m})}) - C$$

$$= (\widetilde{\mathbf{Q}}_{(k'-1)H+h}^{(l)} \widetilde{X}_i^{(l)})^\top (\widetilde{\mathbf{K}}_{(k'-1)H+h}^{(l)} \widetilde{X}_{\xi_m}^{(l)}) - C,$$

where we use Eq. (19) with $k' \neq \kappa$.

To see Eq. (30), we notice that

$$(\widetilde{\mathbf{Q}}_{(\kappa-1)H+h}^{(l)} \widetilde{X}_i^{(l)})^\top (\widetilde{\mathbf{K}}_{(\kappa-1)H+h}^{(l)} \widetilde{X}_j^{(l)}) = (\mathbf{Q}_{\kappa;h}^{(l)} X_{\kappa;i}^{(l)})^\top (\mathbf{K}_{\kappa;h}^{(l)} X_{\kappa;j}^{(l)}) + \alpha_m^\top A_\kappa \alpha_0$$

$$\geq (\mathbf{Q}_{\kappa;h}^{(l)} X_{\kappa;i}^{(l)})^\top (\mathbf{K}_{\kappa;h}^{(l)} X_{\kappa;j'}^{(l)}) + \alpha_m^\top A_\kappa (\alpha_{\iota(j')} + \beta_{\mathcal{E}(v_{j'})}) + C$$

$$= (\widetilde{\mathbf{Q}}_{(\kappa-1)H+h}^{(l)} \widetilde{X}_i^{(l)})^\top (\widetilde{\mathbf{K}}_{(\kappa-1)H+h}^{(l)} \widetilde{X}_{j'}^{(l)}) + C,$$

and

$$(\widetilde{\mathbf{Q}}_{(\kappa-1)H+h}^{(l)} \widetilde{X}_i^{(l)})^\top (\widetilde{\mathbf{K}}_{(\kappa-1)H+h}^{(l)} \widetilde{X}_j^{(l)}) = (\mathbf{Q}_{\kappa;h}^{(l)} X_{\kappa;i}^{(l)})^\top (\mathbf{K}_{\kappa;h}^{(l)} X_{\kappa;j}^{(l)}) + \alpha_m^\top A_\kappa \alpha_0$$

$$\geq (\mathbf{Q}_{\kappa;h}^{(l)} X_{\kappa;i}^{(l)})^\top (\mathbf{K}_{\kappa;h}^{(l)} X_{\kappa;j'}^{(l)}) + \alpha_m^\top A_k \alpha_{\iota(j')} + C$$

$$= (\widetilde{\mathbf{Q}}_{(k-1)H+h}^{(l)} \widetilde{X}_i^{(l)})^\top (\widetilde{\mathbf{K}}_{(k-1)H+h}^{(l)} \widetilde{X}_{j'}^{(l)}) + C,$$

where we use Eq. (20) and Eq. (22).

When $\xi_m < j_1 < j_2$, Eq. (31) follows directly from

$$(\widetilde{\mathbf{Q}}_{(\kappa-1)H+h}^{(l)} \widetilde{X}_i^{(l)})^\top (\widetilde{\mathbf{K}}_{(\kappa-1)H+h}^{(l)} \widetilde{X}_{j_1}^{(l)}) - (\widetilde{\mathbf{Q}}_{(\kappa-1)H+h}^{(l)} \widetilde{X}_i^{(l)})^\top (\widetilde{\mathbf{K}}_{(\kappa-1)H+h}^{(l)} \widetilde{X}_{j_2}^{(l)})$$

$$= (\mathbf{Q}_{\kappa;h}^{(l)} X_{\kappa;i}^{(l)})^\top (\mathbf{K}_{\kappa;h}^{(l)} X_{\kappa;j_1}^{(l)}) + \alpha_m^\top A_k \alpha_m^\top$$

$$\quad - (\mathbf{Q}_{\kappa;h}^{(l)} X_{\kappa;i}^{(l)})^\top (\mathbf{K}_{\kappa;h}^{(l)} X_{\kappa;j_2}^{(l)}) + \alpha_m^\top A_k \alpha_m^\top$$

$$= (\mathbf{Q}_{\kappa;h}^{(l)} Y_{\kappa;i-\xi_m-1+\xi_1}^{(l)})^\top (\mathbf{K}_{\kappa;h}^{(l)} Y_{j_1-\xi_m-1+\xi_1}^{(l)}) - (\mathbf{Q}_{\kappa;h}^{(l)} Y_{i-\xi_m-1+\xi_1}^{(l)})^\top \mathbf{K}_{\kappa;h}^{(l)} Y_{\kappa;j_2-\xi_m-1+\xi_1}^{(l)}).$$

The other cases follow similarly due to Eq. (22).

We have hence confirmed Eq. (29), Eq. (30), Eq. (31), and therefore

$$\frac{\exp\left((\widetilde{\mathbf{Q}}_{(k-1)H+h}^{(l)} \widetilde{X}_i^{(l)})^\top (\widetilde{\mathbf{K}}_{(k-1)H+h}^{(l)} \widetilde{X}_j^{(l)})\right)}{\widetilde{Z}_{(k-1)H+h}^{(l)}} = \begin{cases} \delta_j^{\xi_m}, & k \neq \kappa \\ \dfrac{\exp\left((\mathbf{Q}_{\kappa;h}^{(l)} Y_{\kappa;i-\xi_m-1+\xi_1}^{(l)})^\top (\mathbf{K}_{\kappa;h}^{(l)} Y_j^{(l)})\right)}{\widetilde{Z}_{(k-1)H+h}^{(l)}}, & k = \kappa,\ j < \xi_1 \\ 0, & k = \kappa,\ \xi_1 \leq j \leq \xi_m \\ \dfrac{\exp\left((\mathbf{Q}_{\kappa;h}^{(l)} Y_{\kappa;i-\xi_m-1+\xi_1}^{(l)})^\top (\mathbf{K}_{\kappa;h}^{(l)} Y_{j-\xi_m-1+\xi_1}^{(l)})\right)}{\widetilde{Z}_{(k-1)H+h}^{(l)}}, & k = \kappa,\ j > \xi_m \end{cases}$$

and

$$\widetilde{Z}_{(k-1)H+h}^{(l)} = \sum_{j=1,\ldots,\xi_1-1,\xi_m+1,\ldots,n} \exp\left((\mathbf{Q}_{\kappa;h}^{(l)} Y_{\kappa;i-\xi_m-1+\xi_1}^{(l)})^\top (\mathbf{K}_{\kappa;h}^{(l)} Y_j^{(l)})\right).$$

It follows that

$$X_{\kappa;i}^{(l+1)} = \sum_{j=1}^{\xi_1-1} \frac{\exp\left((\mathbf{Q}_{\kappa;h}^{(l)} Y_{\kappa;i-\xi_m-1+\xi_1}^{(l)})^\top (\mathbf{K}_{\kappa;h}^{(l)} Y_j^{(l)})\right)}{\widetilde{Z}_{(\kappa-1)H+h}^{(l)}} \mathbf{V}_{\kappa;h}^{(l)} Y_j^{(l)}$$

$$\quad + \sum_{j=\xi_m+1}^{i} \frac{\exp\left((\mathbf{Q}_{\kappa;h}^{(l)} Y_{\kappa;i-\xi_m-1+\xi_1}^{(l)})^\top (\mathbf{K}_{\kappa;h}^{(l)} Y_{j-\xi_m-1+\xi_1}^{(l)})\right)}{\widetilde{Z}_{(\kappa-1)H+h}^{(l)}} \mathbf{V}_{\kappa;h}^{(l)} Y_{j-\xi_m-1+\xi_1}^{(l)},$$

$$= Y_{\kappa;i-\xi_m-1+\xi_1}^{(l+1)}$$

$$X_{k';i}^{(l+1)} = X_{k';\xi_m}^{(l)} = 0, \forall k' \neq \kappa.$$

Therefore we establish Eq. (27). This completes the induction.

At the output layer, we have

$$p_{\widetilde{f}}(y|v_1, \ldots, v_n) = \text{Softmax}(\widetilde{\vartheta}(y)^\top \widetilde{X}_n^{(L)})$$

$$= \text{Softmax}(\vartheta(y)^\top Y_{n-\xi_m-1+\xi_1}^{(L)})$$

$$= p_{f_\kappa}(y|u_1, \ldots, u_{n-\xi_m-1+\xi_1}).$$

This establishes the desired Eq. (2). $\qquad\square$

### A.5 Proof of Theorem 4.7

*Proof.* Let $\phi_s, \phi_m, \phi_e$ denote the general-purpose Transformers in Proposition 4.4 (with $K$ experts), 4.2 (with $K = 3$ token spaces), and A.1 (extending to $\mathcal{V}$) respectively. We construct a dummy Transformer $f_d$ that outputs BOS immediately after a token in $\mathcal{A}$. Then we claim that the general-purpose Transformer $\widetilde{\phi}$ defined by

$$\widetilde{\phi}(f_0, f_1, \ldots, f_K) = \phi_m(\phi_s(\phi_e(f_1), \ldots, \phi_e(f_K)), f_d, f_0)$$

achieves the desired property.

Indeed, let $g_1 = \phi_s(\phi_e(f_1), \ldots, \phi_e(f_K))$, by Proposition 4.4, we have

1. **Expert following**: At $t$-th iteration,

$$p_{g_1}\left(\cdot\Big|\text{prompt}\right) \sim p_{f_{a(t)}}\left(\cdot\Big|q|u_{1:i-1}^{(t)}\right),$$

where $q|u_{1:i-1}^{(t)}$ is the token sequence obtained by concatenating the user query $q$ and prior generated part in response $t$: $u_{1:i-1}^{(t)}$.

2. **Regret minimization**:

$$\max_{a^* \in \mathcal{A}} r_0(a^*) - \mathbb{E}[r_0(a^{(T)})] \leq \text{reg}(T).$$

Therefore by Proposition 4.2, we have

$$u_i^{(t)} \sim p_{f_{a(t)}}\left(\cdot\Big|q|u_{1:i-1}^{(t)}\right).$$

It follows that

$$\max_{u^* \in \mathcal{V}^\omega} r(q, u^*) - \mathbb{E}[r(q, u^{(T)})] \leq \lambda + \mathbb{E}_{u \sim f_{k^*}(\cdot|q)}[r(q, u)] - \mathbb{E}_{a^{(T)}}\left[\mathbb{E}_{u^{(T)} \sim f_{a(T)}(\cdot|q)}[r(q, u^{(T)})]\right]$$

$$\leq \lambda + \max_{a^* \in \mathcal{A}} r_0(a^*) - \mathbb{E}[r_0(a^{(T)})]$$

$$\leq \lambda + \text{reg}(T).$$

Finally, $\widetilde{\phi}$ has type $\phi$ of type $(O(K), O(\log(N_{\max})))$ because $\phi_s$ has type $(O(K), O(\log(N_{\max})))$ and $\phi_m, \phi_e$ has type $(O(1), O(\log(N_{\max})))$. This completes the proof. $\qquad\square$

### A.6 Attention Sink Positional Encoding

In this section, we introduce positional encoding mechanisms that induce attention sink behaviors used by Theorem 4.7.

**Lemma A.2** (Attention Sink Positional Encoding, Type 1). *For any $C \in \mathbb{R}_+$, $K, N \in \mathbb{Z}_+$, there exist vectors $\alpha_0, \alpha_1, \ldots, \alpha_N, \beta_1, \ldots, \beta_K \in \mathbb{R}^d$ and matrices $A, A_1, \ldots, A_K \in \mathbb{R}^{d \times d}$ for $d = O(K + \log N)$ such that for any $n \in [N]$ the followings hold:*

*1. For any $k \neq k'$:*

$$\alpha_n^\top A_k(\alpha_n + \beta_{k'}) \geq C + \max\left\{\alpha_n^\top A_k \alpha_n, \max_{0 \leq j < n} \alpha_n^\top A_k \alpha_j, \max_{\substack{1 \leq j < n \\ k'' \in [K]}} \alpha_n^\top A_k(\alpha_j + \beta_{k''})\right\}.$$

2. *For any $k \in [K]$:*
$$\alpha_n^\top A_k \alpha_n = \alpha_n^\top A_k \alpha_0 \geq C + \max\left\{\alpha_n^\top A_k(\alpha_n + \beta_k), \max_{0<j<n} \alpha_n^\top A_k \alpha_j, \max_{\substack{0<j<n \\ k' \in [K]}} \alpha_n^\top A_k(\alpha_j + \beta_{k'})\right\}.$$

3. *For any $k, k' \in [K]$:*
$$(\alpha_n + \beta_{k'})^\top A_k(\alpha_n + \beta_{k'}) \geq C + \max_{0 \leq j \leq n}(\alpha_n + \beta_{k'})^\top A_k \alpha_j.$$

4. *For any $0 < j < n$:*
$$\alpha_n^\top A \alpha_n \geq \alpha_n^\top A(\alpha_n + \beta_k) + C$$
$$\geq C + \max\left\{\alpha_n^\top A \alpha_j, \, \alpha_n^\top A(\alpha_j + \beta_{k'})\right\}, \qquad \forall k, k' \in [K].$$

*Proof.* By Claim A.4, we can find $\gamma_1, \ldots, \gamma_N \in \mathbb{R}^{\bar{d}}$ such that $\bar{d} = O(\log N)$,
$$|\gamma_i^\top \gamma_j| \leq \frac{1}{2}, \qquad \forall i \neq j \in [N], \qquad \text{and} \qquad \gamma_i^\top \gamma_i = 1, \qquad \forall i \in [N].$$

Let $e_1, \ldots, e_K$ be the standard basis of $\mathbb{R}^K$, and let $\mathbf{1}_K$ denote the all-one vector in $\mathbb{R}^K$. We use the block decomposition
$$\mathbb{R}^d = \mathbb{R}^{\bar{d}} \oplus \mathbb{R} \oplus \mathbb{R}^K \oplus \mathbb{R}.$$

Define
$$\alpha_i = \begin{pmatrix} \gamma_i \\ 1 \\ 0 \\ 0 \end{pmatrix}, \qquad \alpha_0 = \begin{pmatrix} 0 \\ 2 \\ 0 \\ 0 \end{pmatrix}, \qquad \beta_k = \begin{pmatrix} 0 \\ 0 \\ e_k \\ 1 \end{pmatrix}.$$

For each $k \in [K]$, define $w_k \in \mathbb{R}^K$ by
$$(w_k)_{k'} = \begin{cases} -C, & k' = k, \\ C, & k' \neq k. \end{cases}$$

Let
$$A_k = \begin{pmatrix} 4C \cdot I_{\bar{d}} & 0 & 0 & 0 \\ 0 & 4C & w_k^\top & 0 \\ 0 & 0 & 0 & 0 \\ 0 & 0 & 0 & 2C \end{pmatrix}, \qquad A = \begin{pmatrix} 4C \cdot I_{\bar{d}} & 0 & 0 & 0 \\ 0 & 0 & -C \cdot \mathbf{1}_K^\top & 0 \\ 0 & 0 & 0 & 0 \\ 0 & 0 & 0 & 0 \end{pmatrix}.$$

The dimension can be bounded by $d = \bar{d} + K + 2 = O(K + \log N)$.

We first verify the properties involving $A_k$. Direct computation gives
$$\alpha_n^\top A_k \alpha_n = \alpha_n^\top A_k \alpha_0 = 8C.$$

Moreover, for $1 \leq j < n$,
$$\alpha_n^\top A_k \alpha_j = 4C\gamma_n^\top \gamma_j + 4C \leq 6C.$$

For any $k' \in [K]$,
$$\alpha_n^\top A_k(\alpha_n + \beta_{k'}) = 8C + (w_k)_{k'} = \begin{cases} 7C, & k' = k, \\ 9C, & k' \neq k. \end{cases}$$

For $1 \leq j < n$ and $k' \in [K]$,
$$\alpha_n^\top A_k(\alpha_j + \beta_{k'}) = 4C\gamma_n^\top \gamma_j + 4C + (w_k)_{k'} \leq 7C.$$

Therefore, if $k' \neq k$, then
$$\alpha_n^\top A_k(\alpha_n + \beta_{k'}) = 9C$$

is at least $C$ larger than all terms in item 1. This proves item 1. Similarly, $\alpha_n^\top A_k \alpha_n = 8C$ is at least $C$ larger than $\alpha_n^\top A_k(\alpha_n + \beta_k) = 7C$, all earlier unmarked scores, and all earlier marked scores. This proves item 2.

For item 3, direct computation gives
$$(\alpha_n + \beta_{k'})^\top A_k(\alpha_n + \beta_{k'}) = 10C + (w_k)_{k'} \geq 9C.$$

On the other hand, for every $0 \leq j \leq n$,

$$(\alpha_n + \beta_{k'})^\top A_k \alpha_j \leq 8C.$$

This proves item 3.

It remains to verify item 4. We have

$$\alpha_n^\top A \alpha_n = 4C, \qquad \alpha_n^\top A(\alpha_n + \beta_k) = 3C.$$

For $0 < j < n$,

$$\alpha_n^\top A \alpha_j = 4C \gamma_n^\top \gamma_j \leq 2C,$$

and

$$\alpha_n^\top A(\alpha_j + \beta_{k'}) = 4C \gamma_n^\top \gamma_j - C \leq C.$$

Thus

$$4C \geq 3C + C, \qquad 3C \geq C + \max\{2C, C\}.$$

This proves item 4 and completes the proof. $\qquad\square$

**Lemma A.3** (Attention Sink Positional Encoding, Type 2). *For any $C \in \mathbb{R}_+$, $K, N \in \mathbb{Z}_+$, there exist vectors $\alpha_1, \ldots, \alpha_N, \beta_0, \ldots, \beta_K \in \mathbb{R}^d$ and matrices $A, A_0, A_1, \ldots, A_K \in \mathbb{R}^{d \times d}$ for $d = O(K + \log N)$ such that for any $n \in [N]$ the followings hold:*

1. *For any $i \geq j_1, j_2$ and $k, k', k'' \in [K]$:*
$$(\alpha_i + \beta_k)^\top A_0(\alpha_{j_1} + \beta_{k'}) = (\alpha_i + \beta_k)^\top A_0(\alpha_{j_2} + \beta_{k''}) \geq (\alpha_i + \beta_k)^\top A_0(\alpha_{j_1} + \beta_0) + C.$$
   *Moreover, for any $i \geq j_1$ and $k \in [K]$:*
$$(\alpha_i + \beta_0)^\top A_0(\alpha_i + \beta_0) \geq (\alpha_i + \beta_0)^\top A_0(\alpha_{j_1} + \beta_k) + C.$$

2. *For any $i > j$ and $k, k' \in [K]$:*
$$\begin{aligned}(\alpha_i + \beta_k)^\top A(\alpha_i + \beta_k) &\geq (\alpha_i + \beta_k)^\top A(\alpha_j + \beta_{k'}) + C \\ &\geq (\alpha_i + \beta_k)^\top A(\alpha_j + \beta_0) + 2C.\end{aligned}$$

3. *For any $i \geq j, j_1$, $k, k', k'' \in [K]$ with $k' \neq k$:*
$$(\alpha_i + \beta_k)^\top A_{k'}(\alpha_j + \beta_0) \geq (\alpha_i + \beta_k)^\top A_{k'}(\alpha_{j_1} + \beta_{k''}) + C.$$
   *Moreover, for any $i \geq j_1$, $k, k', k'' \in [K]$ with $k' \neq k$ and $(j_1, k'') \neq (i, k)$:*
$$(\alpha_i + \beta_k)^\top A_k(\alpha_i + \beta_k) \geq \max\left\{(\alpha_i + \beta_k)^\top A_k(\alpha_{j_1} + \beta_{k''}), (\alpha_i + \beta_k)^\top A_{k'}(\alpha_{j_1} + \beta_0)\right\} + C.$$

*Proof.* By Claim A.4, we can find $\gamma_1, \ldots, \gamma_N \in \mathbb{R}^{\bar{d}}$ such that $\bar{d} = O(\log N)$,

$$|\gamma_i^\top \gamma_j| \leq \frac{1}{2}, \qquad \forall i \neq j \in [N], \qquad \text{and} \qquad \gamma_i^\top \gamma_i = 1, \qquad \forall i \in [N].$$

Let $e_1, \ldots, e_K$ be the standard basis of $\mathbb{R}^K$, and let $\mathbf{1}_K$ denote the all-one vector in $\mathbb{R}^K$. We use the block decomposition

$$\mathbb{R}^d = \mathbb{R}^{\bar{d}} \oplus \mathbb{R}^K \oplus \mathbb{R} \oplus \mathbb{R}.$$

Define

$$\alpha_i = \begin{pmatrix} \gamma_i \\ 0 \\ 0 \\ 0 \end{pmatrix}, \qquad \beta_k = \begin{pmatrix} 0 \\ e_k \\ 1 \\ 0 \end{pmatrix}, \qquad \beta_0 = \begin{pmatrix} 0 \\ 0 \\ 0 \\ 1 \end{pmatrix}.$$

Define

$$A_0 = \begin{pmatrix} 0 & 0 & 0 & 0 \\ 0 & 0 & 0 & 0 \\ 0 & 0 & 2C & 0 \\ 0 & 0 & 0 & C \end{pmatrix}, \qquad A = \begin{pmatrix} 2C \cdot I_{\bar{d}} & 0 & 0 & 0 \\ 0 & 0 & 0 & 0 \\ 0 & 0 & 2C & 0 \\ 0 & 0 & 0 & 0 \end{pmatrix}.$$

For each $k \in [K]$, let $u_k = \mathbf{1}_K - e_k$ and define

$$
A_k = \begin{pmatrix} 2C \cdot I_{\bar{d}} & 0 & 0 & 0 \\ 0 & 6C \cdot e_k e_k^\top & 0 & 5C \cdot u_k \\ 0 & 0 & 0 & 0 \\ 0 & 0 & 0 & 0 \end{pmatrix}.
$$

The dimension can be bounded by $d = \bar{d} + K + 2 = O(K + \log N)$.

We first verify item 1. For any $i, j$ and $k, k' \in [K]$,

$$
(\alpha_i + \beta_k)^\top A_0 (\alpha_j + \beta_{k'}) = 2C,
$$

whereas

$$
(\alpha_i + \beta_k)^\top A_0 (\alpha_j + \beta_0) = 0.
$$

Also,

$$
(\alpha_i + \beta_0)^\top A_0 (\alpha_i + \beta_0) = C, \qquad (\alpha_i + \beta_0)^\top A_0 (\alpha_j + \beta_k) = 0.
$$

This proves item 1.

For item 2, if $i > j$, then

$$
(\alpha_i + \beta_k)^\top A (\alpha_i + \beta_k) = 4C.
$$

Moreover,

$$
(\alpha_i + \beta_k)^\top A (\alpha_j + \beta_{k'}) = 2C \gamma_i^\top \gamma_j + 2C \leq 3C,
$$

and

$$
(\alpha_i + \beta_k)^\top A (\alpha_j + \beta_0) = 2C \gamma_i^\top \gamma_j \leq C.
$$

Therefore,

$$
4C \geq 3C + C, \qquad 3C \geq C + 2C,
$$

which proves item 2.

It remains to verify item 3. Direct computation gives

$$
(\alpha_i + \beta_k)^\top A_{k'} (\alpha_j + \beta_{k''}) = 2C \gamma_i^\top \gamma_j + 6C \cdot \mathbf{1}\{k = k'\} \mathbf{1}\{k'' = k'\},
$$

and

$$
(\alpha_i + \beta_k)^\top A_{k'} (\alpha_j + \beta_0) = 2C \gamma_i^\top \gamma_j + 5C \cdot \mathbf{1}\{k \neq k'\}.
$$

If $k' \neq k$, then

$$
(\alpha_i + \beta_k)^\top A_{k'} (\alpha_j + \beta_0) \geq -C + 5C = 4C,
$$

while

$$
(\alpha_i + \beta_k)^\top A_{k'} (\alpha_{j_1} + \beta_{k''}) \leq C.
$$

Thus the first inequality in item 3 holds.

Finally,

$$
(\alpha_i + \beta_k)^\top A_k (\alpha_i + \beta_k) = 2C + 6C = 8C.
$$

For $(j_1, k'') \neq (i, k)$, either $j_1 \neq i$, in which case

$$
(\alpha_i + \beta_k)^\top A_k (\alpha_{j_1} + \beta_{k''}) \leq C + 6C = 7C,
$$

or $j_1 = i$ and $k'' \neq k$, in which case

$$
(\alpha_i + \beta_k)^\top A_k (\alpha_i + \beta_{k''}) = 2C.
$$

In both cases,

$$
(\alpha_i + \beta_k)^\top A_k (\alpha_{j_1} + \beta_{k''}) \leq 7C.
$$

Moreover, for $k' \neq k$,

$$
(\alpha_i + \beta_k)^\top A_{k'} (\alpha_{j_1} + \beta_0) \leq 2C + 5C = 7C.
$$

Since the matching self-score is $8C$, the second inequality in item 3 follows. This completes the proof. $\qquad \square$

## A.7 TECHNICAL CLAIMS

**Claim A.4** (Johnson-Lindenstrauss Lemma). Given $0 < \varepsilon < 1$, a set $X$ of $N$ points in $\mathbb{R}^n$, and an integer $k > \frac{8(\ln N)}{\varepsilon^2}$, there is a linear map $f : \mathbb{R}^n \to \mathbb{R}^k$ such that

$$(1 - \varepsilon)\|u - v\|^2 \leq \|f(u) - f(v)\|^2 \leq (1 + \varepsilon)\|u - v\|^2$$

holds for all $u, v \in X$.

**Claim A.5** (Berry-Esseen theorem). If $X_1, X_2, \ldots$ are i.i.d. random variables with $\mathbb{E}(X_1) = 0$, $\mathbb{E}(X_1^2) = \sigma^2 > 0$, and $\mathbb{E}(|X_1|^3) = \rho < \infty$, we define

$$Y_n = \frac{X_1 + X_2 + \cdots + X_n}{n}$$

as the sample mean, with $F_n$ the cumulative distribution function of $\frac{Y_n \sqrt{n}}{\sigma}$ and $\Phi$ the cumulative distribution function of the standard normal distribution, then for all $x$ and $n$,

$$|F_n(x) - \Phi(x)| \leq \frac{8\rho}{\sigma^3 \sqrt{n}}.$$

## B    EXPERIMENT DETAILS

### B.1    IMPLEMENTATION DETAILS OF SELF-CORRECTION EXPERIMENTS

The model configurations are detailed in Table 2. Our code is implemented based on `PyTorch` Paszke et al. (2019) and `minGPT`[3]. All the models are trained on one NVIDIA GeForce RTX 2080 Ti GPU with 11GB memory.

| Model | Depth | Heads | Width |
|---|---|---|---|
| GPT-nano | 3 | 3 | 48 |
| GPT-micro | 4 | 4 | 128 |
| GPT-mini | 6 | 6 | 192 |
| Gopher-44M | 8 | 16 | 512 |

Table 2: Model configuration hyperparameters.

Following common practice, the learning rate goes through the warm-up stage in the first 5% of training iterations, and then decays linearly to 0 until training finishes. We set the peak learning rate to $10^{-4}$ and find that all the models are stably trained under this learning rate schedule. We do not apply dropout or weight decay during training. We repeat the experiments three times under different random seeds and report the average accuracy with error bars.

### B.2    PROMPTS FOR SELF-CORRECTION

---

**Initial Problem Solving Prompt**

Solve the following math problem efficiently and clearly. The last line of your response should be of the following format: 'Therefore, the final answer is: `$\boxed{ANSWER}$`. I hope it is correct' (without quotes) where `ANSWER` is just the final number or expression that solves the problem. Think step by step before answering.
{Question}

---

**Correction Prompt**

Your answer is incorrect. Please analyze your solution and identify where you made an error. Then provide a corrected solution that leads to the right answer. The last line of your response should be of the following format: 'Therefore, the final answer is: `$\boxed{ANSWER}$`.'

---

## C    BEST-OF-$n$ WITH IMPERFECT VERIFICATION

We additionally perform experiments to test robustness of best-of-$n$ under imperfect verification. Using Qwen3-32B as an LLM verifier, we evaluated best-of-$n$ on the AIME 24/25 datasets with 3 recent models. As shown in Table 3, best-of-$n$ with an LLM verifier closely matches the performance using an oracle verifier. This evidence supports the practical validity of our theoretical assumptions in Section 5.2.

| Model | Best-of-$n$ (64, oracle) | Best-of-$n$ (64, LLM verifier) |
|---|---|---|
| Qwen3-1.7B | 80.00 | 76.56 |
| Qwen3-4B | 91.67 | 85.20 |
| Llama-3.2-3B-Instruct | 23.33 | 21.45 |

Table 3: Comparison of best-of-64 performance using an oracle verifier vs. Qwen3-32B as an LLM verifier on AIME 24/25. Results are similar across all evaluated models, indicating robustness to imperfect verification.

---

[3]`https://github.com/karpathy/minGPT` (MIT license).

## D LIMITATIONS

Despite these contributions, our work comes with limitations: our construction in Theorem 4.7 only applies to attention-only Transformers and relies on a slightly generalized position encoding method. Relaxing these constraints constitutes interesting problems for future research.

## LARGE LANGUAGE MODELS USAGE DISCLOSURE

LLMs were used only to polish writing.

