# OpenReview forum: "Sample Complexity and Representation Ability of Test-time Scaling Paradigms"
_ICLR.cc/2026/Conference — ICLR 2026 Poster_

### Official Review · Reviewer_YT7w · 2025-10-27

**Soundness:** 3
**Presentation:** 3
**Contribution:** 2
**Rating:** 4
**Confidence:** 4

**Summary:**

This paper explores the theoretical foundations of test-time scaling methods for large language models (LLMs), focusing on the sample efficiency and representational power of strategies like self-consistency, best-of-n, and self-correction. The authors first establish that best-of-n sampling is significantly more sample-efficient than self-consistency for arriving at a correct answer. Secondly, they demonstrate that a Transformer model, when provided with verifier feedback, can simulate online learning over a pool of "expert" models during test time. This allows a single Transformer architecture to solve a variety of tasks without needing to know the specific task in advance, thereby extending the theory of Transformer expressiveness to multi-task scenarios. The paper's theoretical claims are then validated through empirical experiments.

**Strengths:**

* This paper provides a novel conceptual framework for how a single Transformer can achieve multi-task representation.
* Framing self-correction as an online learning problem (i.e., selecting an optimal "expert" ) is an interesting approach.

**Weaknesses:**

* The comparison between best-of-n and self-consistency is unfair. BoN is granted access to a "perfect verifier" while SC is not, making its superior efficiency an expected outcome of having an external signal based not he standard concentration results
* The theoretical model relies on non-standard components, e.g., generalized position encoders. It is unclear if this specific construction applies to the mechanisms of standard Transformers given information beyond positions provided.
* The experiment in 5.1 demonstrates the capacity for self-correction, but it does not validate that this occurs via the specific "expert-selection" mechanism proposed in the theory.

**Questions:**

- Is it possible to establish a sample complexity result of the self-correct method for a comparison with boN and SC to match the result in Table 1?
- While I appreciate the multitask construction run general purpose transformer, it seems an independent result which is used to construct an online learning transformer. This result seems does not directly connect to self correctness but instead in-context RL [1][2], which is widely studied in literature. Could authors clarify more on this?


[1] Lin, Licong, Yu Bai, and Song Mei. "Transformers as decision makers: Provable in-context reinforcement learning via supervised pretraining." arXiv preprint arXiv:2310.08566 (2023).
[2] Lee, Jonathan, et al. "Supervised pretraining can learn in-context reinforcement learning." Advances in Neural Information Processing Systems 36 (2023): 43057-43083.

---

> ### Author Response · Authors · 2025-11-21
>
> Thank you for your insightful and constructive comments. We address each point below.
>
> > The comparison between best-of-n and self-consistency is unfair. BoN is granted access to a "perfect verifier" while SC is not
>
> - Our results do **not** require a perfect reward signal. We only assume that the reward assigns the **highest** value to the correct answer; rewards for incorrect answers may take arbitrary smaller values. This assumption is natural for comparing self-consistency and best-of-$n$: if the model’s probability p as verifier fails to assign the maximal score to the correct solution, then self-consistency will also fail.
> - In many computational problems (e.g., Boolean SAT, integer factorization, graph coloring), *verification* is substantially easier than *generation*, enabling simple oracle verifiers without prior access to the ground-truth solution. Formal proof verification (e.g., via LEAN) is one such example.
> - We additionally performed experiments to test robustness under imperfect verification. Using Qwen3-32B as an LLM verifier, we evaluated best-of-$n$ on the AIME 24/25 datasets with 3 recent models. As shown below, **best-of-$n$ with an LLM verifier closely matches the performance using an oracle verifier**, supporting the practical validity of our theoretical assumptions.
>
>   | Model   | Best-of-$n$ (64, oracle) | Best-of-$n$ (64, LLM verifier) |
>   |--|--|--|
>   | Qwen3-1.7B | 80.00  | 76.56 |
>   | Qwen3-4B   | 91.67  | 85.20 |
>   | Llama-3.2-3B-Instruct | 23.33 | 21.45 |
>
>
> > The theoretical model relies on non-standard components, e.g., generalized position encoders.
>
> - The notion of a ''Generalized Position Encoder'' is motivated by prior work on positional encoding variants in transformers [Liu et al., 2019; Peng et al., 2022; Yang et al., 2023; He et al., 2024; Guan et al., 2024]. For instance, He et al. (2024) introduce a bi-level positional encoding that explicitly uses membership information in sets such as {''.'', ''\n''}. Our construction formalizes and generalizes this class of techniques so that the theoretical arguments can be stated cleanly and proved rigorously.
>
> > The experiment in 5.1 ... does not validate that this occurs via the specific "expert-selection" mechanism proposed in the theory.
>
> - Thank you for raising this important point. Due to the scale and opacity of modern LLMs, it is challenging to determine whether they implement our exact regret-minimization construction from Theorem 4.7.
> - However, emerging evidence shows that LLMs exhibit **ICRL** behaviors that are **consistent with regret minimization**:
>   - Song et al. (2025), Tooliense Team (2025), and Monea et al. (2025) demonstrate that LLMs can perform *online adaptation* driven by feedback signals.
>   - Qu et al. (2025) further show that fine-tuning can amplify these abilities, improving test-time decision making.
> - These works suggest that **LLMs can indeed implement mechanisms closely aligned with what our theory analyzes**, even if not in the exact constructive form used for the proof.
>
> > Is it possible to establish a sample complexity result of the self-correct method for a comparison with boN and SC to match the result in Table 1?
>
> - Thank you for the suggestion. Our Theorem 4.7 is a **representation result**, not a sample-complexity bound, and therefore does not directly compare with BoN. BoN’s sample complexity depends on a single margin parameter $\Delta$, while that of SC depends on **additional factors** such as expert diversity, online learning dynamics, and convergence rates. As a result, the two cannot be placed into a single-parameter comparison.
>
> > it seems an independent result which is used to construct an online learning transformer. This result seems does not directly connect to self correctness but instead in-context RL [1][2], which is widely studied in literature.
>
> - Self-correction can be viewed as a **specialized and more challenging form of in-context RL**, where the action space consists of **entire response sequences** of variable length, and the state is the **full contextual history**, not a fixed-dimensional vector. In contrast, prior ICRL theory (e.g., [1][2]) typically assumes **finite-dimensional action spaces** $\mathbb{R}^d$.  Our analysis therefore extends these ideas to a significantly richer setting and focuses specifically on the **self-correction mechanism**, rather than the general ICRL problem.
>
> References:
>
> He, Zhenyu, et al. "Two stones hit one bird: Bilevel positional encoding for better length extrapolation." ICML (2024).
>
> Yang, Wentao, et al. "Read ten lines at one glance: line-aware semi-autoregressive Transformer for multi-line handwritten mathematical expression recognition." Proceedings of the 31st ACM International Conference on Multimedia. 2023.
>
> Guan, Tongkun, et al. "Posformer: recognizing complex handwritten mathematical expression with position forest transformer." European Conference on Computer Vision. Cham: Springer Nature Switzerland, 2024.

---

### Official Review · Reviewer_jvdL · 2025-10-31

**Soundness:** 2
**Presentation:** 3
**Contribution:** 2
**Rating:** 2
**Confidence:** 4

**Summary:**

This paper theoretically investigates the sample complexity and representation ability of test-time scaling paradigms for Large Language Models (LLMs), such as self-consistency, best-of-n, and self-correction.

This paper should be rejected because (1) The paper employs complex and abstract theoretical concepts that are either disconnected from the subsequent proposed method, (2) The paper's structure is fundamentally fragmented, (3) The central Self-Correction method's theoretical framework relies on the existence of an accruate verifier with feedback, which is unrealistic and severe limitation in standard Test-Time Scaling (TTS) paradigms. Seed detailed comments below.

**Strengths:**

1.  The paper provides a novel theoretical framework to analyze the sample complexity of Test-Time Scaling (TTS) methods, establishing a clear separation result between self-consistency ($\Theta(1/\Delta^2)$) and best-of-n ($\Theta(1/\Delta)$).

2.  It offers a new perspective on self-correction, proving its representational power to enable a single Transformer to simulate online learning over a pool of experts (Bandit problem) at test time, thus extending the theory of Transformers from single-task to multi-task settings.

**Weaknesses:**

1.  The paper lacks a unified motivation, splitting into two seemingly disconnected parts: the sample complexity analysis of repeated sampling methods (self-consistency and best-of-n) and the theoretical analysis of Self-Correction with Verifier Feedback. The connection between the two main results is not clearly established.

2.  The analysis of Self-Correction with Verifier Feedback relies on the existence of an _accurate_ verifier, which is generally not a realistic assumption for a test-time scaling paradigm. Specifically, the modeling of the LLM inference process as a Bandit problem, where a General-Purpose Transformer (conceptually similar to MoE) learns to select an expert based on verifier feedback, is novel but appears disconnected from practical Test-Time Scaling (TTS) scenarios.

3.  The paper's structure is confusing. The Introduction elaborates on the implementation details of the "General-Purpose Transformer," while the dedicated Method section provides minimal subsequent discussion. This makes the Introduction overly dense and difficult to follow. The authors should prioritize an intuitive understanding and the main "takeaway" conclusions of the General-Purpose Transformer concept in the Introduction, moving the implementation specifics to the Method section or an Appendix.

4.  The reasoning in the claim "We illustrated this mechanism in Figure 2. As a result, the action sequence achieves $o(1)$ regret and the response sequence is generated from the corresponding expert selected by the latest action. Therefore, the response sequence also achieves regret $o(1)$" is unclear. It is not immediately obvious how merely illustrating a mechanism in Figure 2 leads to the conclusion of $o(1)$ regret for the action sequence.

5.  The Generalized Position Encoder introduced in the Preliminary section appears to be completely irrelevant to the rest of the paper's analysis and discussion.

**Questions:**

- What is the relation between the sample complexity proof and the proposed method?
- How can one obtain a oracle verifier with feedback in practice?

---

> ### Author Response · Authors · 2025-11-21
>
> We address each point below.
>
> > The reasoning in the claim "We illustrated this mechanism in Figure 2. As a result, the action sequence achieves  regret and the response sequence is generated from the corresponding expert selected by the latest action. Therefore, the response sequence also achieves regret " is unclear. It is not immediately obvious how merely illustrating a mechanism in Figure 2 leads to the conclusion of  regret for the action sequence.
>
> - The quoted text **does not appear in our ICLR submission**.
>
> > What is the relation between the sample complexity proof and the proposed method?
>
> - To clarify: **our work does not produce a new algorithmic method**. Instead, we provide:
>   1. **Sample-complexity separation results** showing regimes where self-correction outperforms self-consistency;
>   2. **Representation-theoretic constructions** demonstrating that transformers can implement self-correction mechanisms; and
>   3. **Empirical validations** confirming that modern models exhibit the behaviors predicted by our theory.
>
>
> > The Generalized Position Encoder introduced in the Preliminary section appears to be completely irrelevant to the rest of the paper's analysis and discussion.
>
> - The Generalized Position Encoder is a **core technical component** used in our construction for Theorem 4.7. It is grounded in prior transformer architecture literature [Liu et al., 2019; Peng et al., 2022; Yang et al., 2023; He et al., 2024; Guan et al., 2024] and enables our proof to cleanly encode the structural information required for the self-correction mechanism. While it is not used in experiments, it is necessary for the theoretical representation result.
>
>
> > How can one obtain a oracle verifier with feedback in practice?
> - Our results do **not** require an oracle verifier. We only assume that the reward assigns the **highest** value to the correct answer; rewards for incorrect answers may take arbitrary smaller values. This assumption is natural for comparing self-consistency and best-of-$n$: if the model’s probability p as verifier fails to assign the maximal score to the correct solution, then self-consistency will also fail.
> - In many computational problems (e.g., Boolean SAT, integer factorization, graph coloring), *verification* is substantially easier than *generation*, enabling simple oracle verifiers without prior access to the ground-truth solution. Formal proof verification (e.g., via LEAN) is one such example.
> Huang et al. shows that a carefully-designed inference framework on frontier LLMs can lead to a highly robust verifier even for challenging problems on International Mathematical Olympiad (IMO). While there is no explicit guarantee that the verification or feedback is perfect, it is accurate enough to lead to gold-medal-level performance on IMO.
>
> References:
>
> He, Zhenyu, et al. "Two stones hit one bird: Bilevel positional encoding for better length extrapolation." ICML (2024).
>
> Yang, Wentao, et al. "Read ten lines at one glance: line-aware semi-autoregressive Transformer for multi-line handwritten mathematical expression recognition." Proceedings of the 31st ACM International Conference on Multimedia. 2023.
>
> Guan, Tongkun, et al. "Posformer: recognizing complex handwritten mathematical expression with position forest transformer." European Conference on Computer Vision. Cham: Springer Nature Switzerland, 2024.
>
> Peng, Han, et al. "Rethinking positional encoding in tree transformer for code representation." Proceedings of the 2022 conference on empirical methods in natural language processing. 2022.
>
> He, Haoyu, et al. "HDT: Hierarchical document transformer." Conference on Language Modeling (2024).
>
> Liu, Yang, and Mirella Lapata. "Hierarchical transformers for multi-document summarization." arXiv preprint arXiv:1905.13164 (2019).
>
> Huang, Yichen, and Lin F. Yang. "Winning Gold at IMO 2025 with a Model-Agnostic Verification-and-Refinement Pipeline." arXiv preprint arXiv:2507.15855 (2025).

---

### Official Review · Reviewer_9kos · 2025-10-31

**Soundness:** 3
**Presentation:** 3
**Contribution:** 2
**Rating:** 6
**Confidence:** 4

**Summary:**

This paper analyzes the test‑time scaling of LLMs along two directions. First, it proves a sample‑complexity separation between repeated‑sampling strategies. Self‑consistency needs $\Theta(1/\Delta^2)$ samples while best‑of‑n needs $\Theta(1/\Delta)$, where $\Delta$ is the probability gap between the most likely correct answer and the next alternative. Second, it develops a general‑purpose expressiveness framework, showing that a single wider Transformer equipped with verifier feedback can implement online learning over a pool of expert Transformers at inference time, yielding simple‑regret $\mathrm{reg}(T)$ and final reward within $\lambda+\mathrm{reg}(T)$ of optimal. Experiments also illustrate that self‑correction can boost accuracy on a synthetic task and that best‑of‑n with a verifier can outperform self‑consistency on AIME’24/’25 even with far fewer samples.

**Strengths:**

- The paper connects strands across CoT scaling and verification and makes a clear theoretical contribution on sampling and self‑correction.
- The paper has good technical depth and the mathematical statements/proofs are rigorous with matching upper/lower bounds, together with complementary experiments.
- It's also interesting to have that general‑purpose Transformer constructions manage to route to the correct expert in far less than $K$ trials, which is equivalent to brute‑force trials.

**Weaknesses:**

- The separation results assume a perfect reward for best‑of‑n, the theory does not capture the settings with noisy/imperfect verification.
- The unified construction of transformer using experts is already engineered to convey the claim that transformer does online learning over a pool of experts with verification, so the conclusion feels built‑in. If it was the other way around (i.e., inductive bias of trained transformer on forming experts), the story would be more convincing.
- It would be good to have confidence intervals on AIME accuracies with only 4 samples.
- A full section is devoted to Section 3, yet the technical novelty here is limited as the separation bounds follow directly from known approximation results (w.r.t $\ell_1, \ell_2$, etc.) for multinomial distribution.

**Questions:**

- It's not clearly specified what kind of verification feedback is obtained from Qwen3-32B model and how many steps of verification are performed on AIME benchmarks.
- The expressiveness results depend on an attention‑precision assumption, could you justify why this is necessary?
- The expressiveness proof depends on $\epsilon-$thresholded attention assumption, which causes sparsity. Could you justify the necessity of this assumption?
- I also want to bring to authors' attention that there's a previous work analyzing 3-SAT problems for self-correction, "Learning to Self-Correct through Chain-of-Thought Verification", as the relation to that work is not discussed.

I don't have any more questions, please see weaknesses.

---

> ### Author Response · Authors · 2025-11-21
>
> Thank you for your insightful and constructive comments. We address each point below.
>
> > The separation results assume a perfect reward for best‑of‑n, the theory does not capture the settings with noisy/imperfect verification.
>
> - Our results do **not** require a perfect reward signal. We only assume that the reward assigns the **highest** value to the correct answer; rewards for incorrect answers may take arbitrary smaller values. This assumption is natural for comparing self-consistency and best-of-$n$: if the model’s probability p as verifier fails to assign the maximal score to the correct solution, then self-consistency will also fail.
> - In many computational problems (e.g., Boolean SAT, integer factorization, graph coloring), *verification* is substantially easier than *generation*, enabling simple oracle verifiers without prior access to the ground-truth solution. Formal proof verification (e.g., via LEAN) is one such example.
> - We additionally performed experiments to test robustness under imperfect verification. Using Qwen3-32B as an LLM verifier, we evaluated best-of-$n$ on the AIME 24/25 datasets with 3 recent models. As shown below, **best-of-$n$ with an LLM verifier closely matches the performance using an oracle verifier**, supporting the practical validity of our theoretical assumptions.
>
>   | Model                   | Best-of-$n$ (64, oracle) | Best-of-$n$ (64, LLM verifier) |
>   |--|--|--|
>   | Qwen3-1.7B    | 80.00                     | 76.56                           |
>   | Qwen3-4B | 91.67                     | 85.20                           |
>   | Llama-3.2-3B-Instruct   | 23.33                     | 21.45                           |
>
> > The unified construction of transformer using experts is already engineered to convey the claim that transformer does online learning over a pool of experts with verification, so the conclusion feels built‑in.
>
> - Mixture-of-expert construction of transformers is not an engineered way to convey the claim but a successful design for modern LLMs [Liu et al., 2024]. We agree that understanding the inductive bias of trained transformers toward expert-like representations is an interesting direction, but it is outside the scope of the present work.
>
> > confidence intervals on AIME accuracies with only 4 samples.
>
> - Thank you for the suggestion. The standard deviations of the best-of-4 accuracies are 3.4%, 2.8%, 2.2% for Qwen3-1.7B, Qwen3-4B, and Llama-3.2-3B-Instruct, respectively.
>
> > what kind of verification feedback is obtained from Qwen3-32B model and how many steps of verification are performed on AIME benchmarks.
>
> - Thank you for your careful review. To obtain the verification feedback, we provide the verifier with the problem statement and the current model-generated solution. The verifier is instructed to determine correctness and provide a binary output (correct/incorrect).  For reference, the detailed prompt for feedback generation is
>   ```
>   Problem: {problem}
>   Solution: {solution}
>
>   Is this solution correct? Answer only YES or NO.
>   ```
>
> - Regarding the number of steps: We set the inference budget to 4 samples (iterations) for the self-correction method, as noted in Table 1. This means the model performs up to 4 cycles of generation and verification to refine its answer. In each iteration, the feedback is obtained from a single-turn interaction with the Qwen3-32B model as described above.
>
> > The expressiveness results depend on an attention‑precision assumption, could you justify why this is necessary?
>
> - The assumption is **not fundamentally necessary**; it is used purely to simplify the proof by removing negligible error terms in attention. Without it, each layer’s embedding would carry an additional $O(\epsilon)$ perturbation.
> - This type of precision assumption — i.e., outputs of arithmetic operations are rounded to the nearest representable floating-point value – is **standard in ML representation-theoretic analyses** [Liu et al., 2023; Li et al., 2024].
>
>
> > The expressiveness proof depends on thresholded attention assumption, which causes sparsity. Could you justify the necessity of this assumption?
>
> - When sequence length scales, some degree of **attention sparsity is necessary**: without focusing attention selectively, the model would dilute probability mass over the entire context, reducing its ability to latch onto key information.
>
> > there's a previous work analyzing 3-SAT problems for self-correction, "Learning to Self-Correct through Chain-of-Thought Verification"
>
> - Thank you for bringing this to our attention. We will incorporate a discussion of this related work in the final version.
>
> References:
>
> Liu, Aixin, et al. "Deepseek-v2: A strong, economical, and efficient mixture-of-experts language model." arXiv preprint arXiv:2405.04434 (2024).
>
> Liu, Bingbin, et al. "Transformers learn shortcuts to automata." ICLR (2023).
>
> Li, Zhiyuan, et al. "Chain of thought empowers transformers to solve inherently serial problems." ICLR (2024).

---

### Official Review · Reviewer_zTQW · 2025-11-02

**Soundness:** 3
**Presentation:** 3
**Contribution:** 3
**Rating:** 6
**Confidence:** 2

**Summary:**

This work aims to provide a theoretical foundation for several popular test-time scaling paradigms. The paper's contributions are twofold:

Separation of Sample Complexity: The paper provides the first rigorous theoretical separation between two repeated sampling strategies: self-consistency and best-of-n (BofN). The authors prove that to achieve the same success probability, self-consistency requires a sample complexity of $\Theta(1/\Delta^2)$, whereas BofN only requires $\Theta(1/\Delta)$. Here, $\Delta$ represents the probability gap between the correct answer and the second-most-likely answer. This result theoretically explains why BofN (when equipped with a good verifier) is generally more sample-efficient than self-consistency.

Representation Ability of Self-Correction: The paper then analyzes the representation ability of self-correction with verifier feedback. The authors introduce the concept of a "General-Purpose Transformer" and prove that a Transformer architecture can be constructed to simulate an online learning algorithm over a pool of "experts" at test-time. Specifically, this Transformer can leverage reward feedback from a verifier to adaptively select experts and minimize regret, enabling it to solve multi-task problems without prior knowledge. This extends the known representation theory of Transformers from single-task to multi-task settings.

**Strengths:**

- This work provides a solid theoretical basis for two widely-used but poorly-understood practical heuristics (BofN vs. Self-consistency). The $\Theta(1/\Delta)$ vs. $\Theta(1/\Delta^2)$ separation result is clear, important, and appears fundamental.

- The framework of a "General-Purpose Transformer" and "test-time online learning" is a novel perspective. The proof that a Transformer architecture can (by construction) implement regret-minimizing online learning is a significant extension of Transformer representation theory, moving beyond standard "universal approximator" or "Turing complete" claims.

- The synthetic experiments (on the 3-SAT task) are cleverly designed and successfully demonstrate both (1) the theoretical sample complexity separation and (2) the representation ability of the model to improve its performance via self-correction, validating the paper's core theoretical

**Weaknesses:**

- The theoretical construction of the "General-Purpose Transformer" (Propositions 4.2, 4.4) appears highly complex and relies on a specific "Generalized Position Encoder" (Definition 2.2) and attention sink techniques. This feels more like an existence proof (i.e., "we can construct a Transformer that does this") rather than an explanation of how existing LLMs might learn this behavior through standard pre-training.

- The proof of self-correction's representation ability relies on a non-standard, more powerful "Generalized Position Encoder" that has access to set membership information of preceding tokens. This limits the applicability of this theoretical proof to specific (and perhaps not yet existing) architectures.

**Questions:**

The experiments in Section 5.1 show that larger models (GPT-mini, Gopher-44M) achieve near-perfect accuracy with self-correction feedback. Does this imply that these models are truly simulating the "online learning over an expert pool" described in Theorem 4.7? Or is it more likely that they are simply learning a much stronger, but simpler, heuristic (e.g., "if the answer is 'no', flip the string")? How do the authors view the gap between this complex theoretical construction and the observed empirical result?

---

> ### Author Response · Authors · 2025-11-21
>
> Thank you for your insightful and constructive comments. We address each point below.
>
>
> > The theoretical construction of the "General-Purpose Transformer" (Propositions 4.2, 4.4) appears highly complex and relies on a specific "Generalized Position Encoder" (Definition 2.2) and attention sink techniques. This feels more like an existence proof (i.e., "we can construct a Transformer that does this") rather than an explanation of how existing LLMs might learn this behavior through standard pre-training.
> - We agree that our theoretical framework presents an existence proof. However, our empirical results demonstrate that practical transformers *without* curated generalized positional encodings still exhibit the desired self-correction behavior.
> - Our approach follows a well-established paradigm in the community: developing rigorous theoretical results for a restricted architectural class and then validating that the resulting insights generalize empirically. For example, Feng et al. (NeurIPS 2023, oral) analyze a model class with a specific absolute positional encoding but show experimentally that the theoretical insights also hold for modern relative positional encoding schemes.
>
>
> > The proof of self-correction's representation ability relies on a non-standard, more powerful "Generalized Position Encoder" that has access to set membership information of preceding tokens.
>
> - The notion of a ''Generalized Position Encoder'' is motivated by prior work on positional encoding variants in transformers [Liu et al., 2019; Peng et al., 2022; Yang et al., 2023; He et al., 2024; Guan et al., 2024]. For instance, He et al. (2024) introduce a bi-level positional encoding that explicitly uses membership information in sets such as {''.'', ''\n''}. Our construction formalizes and generalizes this class of techniques so that the theoretical arguments can be stated cleanly and proved rigorously.
>
>
> > The experiments in Section 5.1 show that larger models (GPT-mini, Gopher-44M) achieve near-perfect accuracy with self-correction feedback. Does this imply that these models are truly simulating the "online learning over an expert pool" described in Theorem 4.7? Or is it more likely that they are simply learning a much stronger, but simpler, heuristic (e.g., "if the answer is 'no', flip the string")? How do the authors view the gap between this complex theoretical construction and the observed empirical result?
>
> - This is an excellent point! Our theoretical result offers a potential mechanism for self-correction, but it is also possible that the model implements different correction strategies in specific applications: like the example in your question. Determining the exact model behavior requires mechanistic interpretability insights into this process in real LLMs and is an open challenge.
> - That being said, the empirical observation indicates the larger models are successful precisely because they have the expressiveness to execute this ''test-time online learning'' (maybe in a simpler way) while the smaller models do not. This still supports our paper's central claim that self-correction is a powerful mechanism that requires sufficient model capacity to implement effectively.
>
>
> References:
>
> Feng, Guhao, et al. "Towards revealing the mystery behind chain of thought: a theoretical perspective." Advances in Neural Information Processing Systems 36 (2023): 70757-70798.
>
> He, Zhenyu, et al. "Two stones hit one bird: Bilevel positional encoding for better length extrapolation." ICML (2024).
>
> Yang, Wentao, et al. "Read ten lines at one glance: line-aware semi-autoregressive Transformer for multi-line handwritten mathematical expression recognition." Proceedings of the 31st ACM International Conference on Multimedia. 2023.
>
> Guan, Tongkun, et al. "Posformer: recognizing complex handwritten mathematical expression with position forest transformer." European Conference on Computer Vision. Cham: Springer Nature Switzerland, 2024.
>
> Peng, Han, et al. "Rethinking positional encoding in tree transformer for code representation." Proceedings of the 2022 conference on empirical methods in natural language processing. 2022.
>
> He, Haoyu, et al. "HDT: Hierarchical document transformer." Conference on Language Modeling (2024).
>
> Liu, Yang, and Mirella Lapata. "Hierarchical transformers for multi-document summarization." arXiv preprint arXiv:1905.13164 (2019).

---

### Meta-Review · Area_Chair_oMPn · 2026-01-07

**Summary:**

The reviewers thought the fact that the paper provides some theoretical results for test-time scaling in Large Language Models, establishing a sample complexity separation between self-consistency and best-of-n is interesting. Two reviewers praised the framework, which proves that Transformers can simulate online learning to perform self-correction. However, the reviewers raised significant concerns regarding the theoretical assumptions which according to some reviewers was more an existence proof than explaining model behavior. One reviewer criticized the comparison between strategies as unfair due to unequal access to verifier signals. One reviewer was flagged by the authors for citing text not present in the submission, suggesting the review might be based on a different version of the manuscript. I do think the authors response for the most part addressed valid concerns and the paper will be an interesting addition.

**Reviewer Concerns:**

The authors addressed most issues but did acknowledge that mapping their theoretical construction to exact model mechanisms remains an open challenge, they cited emerging evidence in In-Context Reinforcement Learning to bridge this gap.

**Reviewer Scores:**

Reviewer zTQW (Current: 6): Likely to maintain 6 or increase to 7. The authors acknowledged the "existence proof" limitation within standard theoretical practices.

Reviewer 9kos (Current: 6): Likely to increase to 7. The authors directly provided the requested confidence intervals, missing references, and robust experimental evidence regarding imperfect verifiers.


Reviewer YT7w (Current: 4): Likely to stay the same or increase to 5. The authors provided a detailed defense of their comparison methodology and linked their results to In-Context RL literatures.

Reviewer jvdL (Current: 2): Unlikely to increase. The authors pointed that this reviewer critiqued text not present in the submission, suggesting the review should be discounted by the Area Chair. While this point is valid I don’t think the comments can be fully discounted.

---

### Decision · Program_Chairs · 2026-01-26

Accept (Poster)